# Pancakes: Consistent Multi-Protocol Image Segmentation Across Biomedical Domains

**Marianne Rakic**
MIT CSAIL, MGH
mrakic@mit.edu

**Siyu Gai**
MIT CSAIL, MGH

**Etienne Chollet**
MIT CSAIL, MGH

**John V. Guttag**
MIT CSAIL

**Adrian V. Dalca**
MIT CSAIL, HMS, MGH

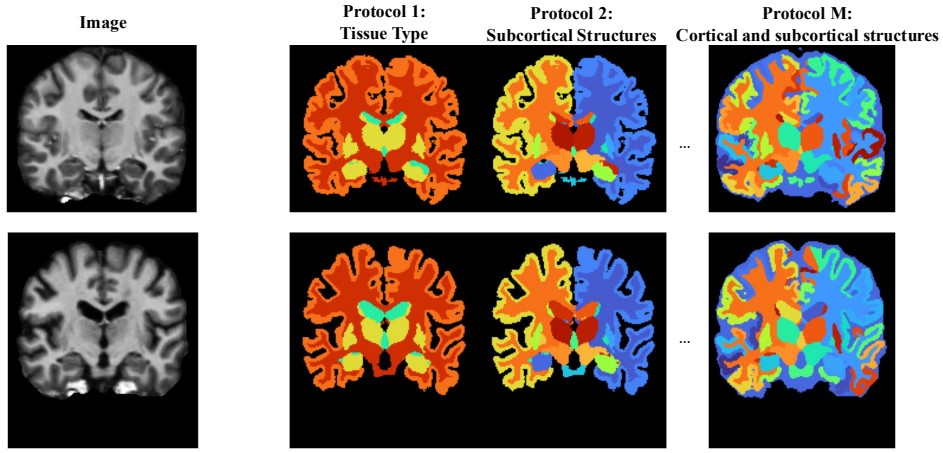

Figure 1: **Examples of different expert provided protocols for brain MRI.** Any biomedical image can be segmented in many different ways. For example, protocol 1 here corresponds to a coarse-grained categorization of tissue types. Colors correspond to distinct ROIs (the choice of colors is arbitrary). Typical neural networks follow a *fixed* protocol, specified explicitly or implicitly by the user.

## Abstract

A single biomedical image can be meaningfully segmented in multiple ways, depending on the desired application. For instance, a brain MRI can be segmented according to tissue types, vascular territories, broad anatomical regions, fine-grained anatomy, or pathology, etc. Existing automatic segmentation models typically either (1) support only a single protocol – the one they were trained on – or (2) require labor-intensive manual prompting to specify the desired segmentation. We introduce Pancakes, a framework that, given a new image from a previously unseen domain, automatically generates multi-label segmentation maps for *multiple* plausible protocols, while maintaining semantic consistency across related images. Pancakes introduces a new problem formulation that is not currently attainable by existing foundation models. In a series of experiments on seven held-out datasets, we demonstrate that our model can significantly outperform existing foundation models in producing several plausible whole-image segmentations, that are semantically coherent across images.

39th Conference on Neural Information Processing Systems (NeurIPS 2025).

# 1 Introduction

There are many ways to segment a biomedical image. Depending on their goals, clinicians or biomedical researchers employ a specific segmentation *protocol*, which defines the regions of interest (ROIs) to be segmented (Figure 1). This could involve, for example, segmenting major anatomical classes, granular anatomical regions, diffuse tissue types, systemic structures like vessels or nerves, pathologies, or functional areas [6, 28, 45, 63, 80, 119].

Existing learning-based biomedical image segmentation tools require specification of a segmentation protocol [16, 58, 116, 123, 130]. Fully-automated models are trained to segment an image using image-segmentation training pairs, which implicitly define the protocol [18, 48, 99]. Recent in-context or few-shot models also use image-segmentation pairs, but take them *as input* to specify the desired segmentation protocol for a target image [8, 16, 96, 117]. Interactive models rely on user interactions to indicate the desired ROIs [58, 116, 122, 123]. In all of these strategies, the desired segmentation protocol is specified by the user (interactively or by example). This is a substantial burden on biomedical researchers, who commonly need to segment a new biomedical image with a potentially new segmentation protocol [4, 16].

Our goal is to support a *new capability*: enabling biomedical researchers and clinicians to explore a diverse set of plausible, semantically consistent segmentations for a previously unseen collection of images. We propose a fundamentally new approach to segmenting a new biomedical dataset. Instead of requiring the user to specify the protocol for a new task, our method, Pancakes, produces segmentation maps for multiple plausible protocols simultaneously, each consistent across images. After Pancakes has generated the label maps, a researcher or clinician can select which of the proposed protocols best aligns with their intended downstream use (e.g., anatomical volume analysis). We envision at least two broad classes of use:

**(1) Rapid segmentation for new protocols.** New protocols are frequently introduced [1, 36, 85], and there is a need to produce corresponding segmentations. If a scientist has a particular protocol in mind, but there are no existing tools for segmenting it, they can choose the protocol from Pancakes that best aligns with their intended use.

**(2) Exploratory population analysis.** Pancakes will support users in discovering or selecting segmentation strategies appropriate for their scientific or clinical questions. For example, a clinical scientist who studies how anatomy relates to some outcome (e.g., progression of a disease) or predictor (e.g., genetics) can use Pancakes to quickly extract segmentations of multiple candidate anatomical regions that have never been segmented, compute their volumes, and test correlations with clinical outcomes thereby identifying promising candidate regions.

Pancakes takes an image as input and produces a *distribution* over segmentation protocols. It then uses a segmentation sampling mechanism to produce several complete, multi-label segmentation maps from diverse protocols for that image. Importantly, within a chosen protocol, segmentation maps are semantically consistent across subjects – a specific label denotes the same anatomical structure in every image of the collection.

In a series of experiments, we demonstrate that our model can significantly outperform baselines in producing several plausible whole-image segmentations that are semantically coherent across images. We show that Pancakes outperforms foundation segmentation models by a wide margin on seven held-out datasets.

# 2 Related work

**Single-protocol biomedical image segmentation.** Most existing biomedical image segmentation models [18, 48, 99] are by design constrained to a specific biomedical domain and image type. For example, some models specialize in segmenting images of brains [13, 28, 38, 77, 92], hearts [112, 134], or eye vessels [50, 70, 100]. Each model learns a specific segmentation protocol defined by the image-segmentation pairs used during training.

**Universal segmentation.** Recent *universal* models can each segment a wide variety of structures across biomedical domains. They are trained jointly on large data collections containing diverse

| Image Set to segment | SAM setting 1 | SAM setting 2 | UnSAM |
| --- | --- | --- | --- |

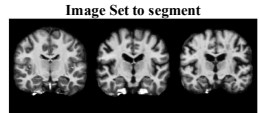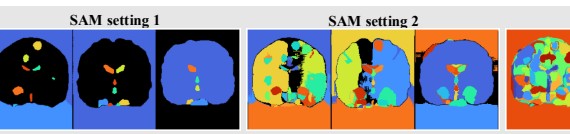

Figure 2: **Current automatic segmentation foundation models produce inconsistent segmentations.** Given a set of similar images to segment (left), automatic foundation models can fully parcellate each image, but the obtained segmentations are not semantically consistent across images. Even in rare cases where the same structure is labeled on two images, the *label index* (color-coded here) is usually inconsistent. There is then no clear mapping between the segmentations from the two samples, making it difficult for biomedical clinicians and researchers to use the results.

structures and image types, both for natural [8, 58, 97, 115, 117] and medical imaging [16, 44, 96, 123, 124, 130, 131]. Some methods generalize to new tasks by enabling a condition, or prompt, as input. This conditioning could involve example image-segmentation pairs [8, 16, 30, 96, 115, 117, 122], user interactions such as bounding boxes, clicks, or scribbles [58, 76, 97, 122–124, 135], or even text [44, 130, 131, 135]. Providing this conditioning is labor-intensive, especially when tackling a new segmentation task involving a large collection of images.

We also train Pancakes using large image collections, to generalize to new domains. However, we avoid the need for the user to laboriously prescribe the protocol and instead automatically estimate several segmentation maps from several plausible protocols.

Some universal models can completely partition an image from a new biomedical domain into multiple labels [58, 68, 116]. As we show in our experiments and illustrate in Figure 2, these segmentation maps are semantically inconsistent across subjects, with the same label having different meanings across images. In the rare case when the same structure is segmented in two images, the assigned label index is usually different, and establishing the correspondence is non-trivial.

**Multi-protocol segmentation.** Motivated by the fact that many objects in an image can be divided into subparts, some methods produce labels from a hierarchy of protocols in an image [24, 88, 114, 125]. This restricts the types of segmentations produced to fixed protocols that are inherently hierarchical. The methods are trained on limited domains or specialized to natural images. In either case, they require the hierarchy to be explicitly provided, which makes them less broadly applicable. In contrast, our approach produces label maps from multiple protocols that need not be hierarchically related, while generalizing to unseen structures.

**Ambiguity and uncertainty.** Even within a well-defined protocol, many segmentation tasks and biomedical images involve substantial ambiguity. This can be caused by problems with the image acquisition (e.g., noise or low contrast), ambiguous definitions of the desired region, or the downstream goals following the segmentation step. Recent models capture variability among manual raters [96, 102, 111], often by aggregating multiple predictions for a given structure or protocol to obtain an uncertainty estimate [23]. In our work, we jointly capture the ambiguity of the possible protocol and the inherent ambiguity in the image, but focus on the ability of a single framework to produce segmentations that represent different protocols consistently across scans.

**Deep-learning and sampling mechanisms.** Deep-learning segmentation frameworks that produce different outputs for a given input use an implicit or explicit mechanism to sample different solutions, such as variational autoencoders [10, 59, 60], diffusion models [95, 120, 121, 126], multivariate Gaussian [86], or in-context stochastic models [96]. We build on these methods, and propose a new mechanism to sample different segmentation protocols that are varied but consistent among images from the same domain.

# 3 Method

Given an image $x$, we let $y_m$ be a multi-label segmentation map for a specific protocol $m$ composed of non-overlapping labels. Typically, a segmentation model $g_\theta(x) = \hat{y}_m$ follows a fixed predefined protocol $m$.

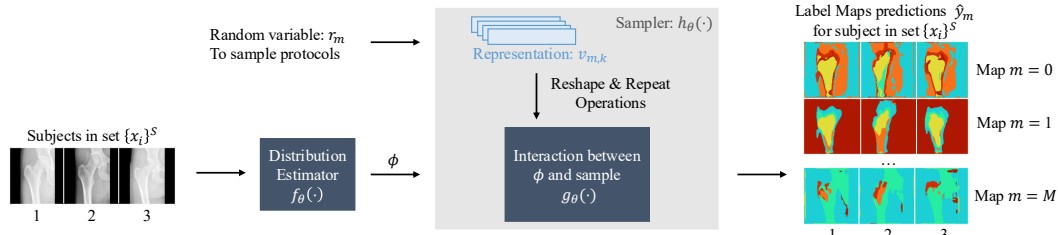

Figure 3: **Method Schematic.** To produce multiple consistent label maps for an image set $\{x_s\}$, we first estimate the distribution parameters $\phi$ via $f_{\theta_f}(x)$. We then sample from the distribution with parameters $\phi$ through the random variable $r_m$: $h_{\theta_h}(r_m, \phi) = \hat{y}_m$.

Instead, we design a framework that can produce a set of label maps $\{y_m\}_{m=1}^{M}$ for $M$ different protocols, summarized in Figure 3. We estimate high-dimensional parameters $\phi$ of a distribution $p(y_m; \phi|x)$ over segmentation maps $y_m$ spanning different segmentation protocols $m$, using the function $f_{\theta_f}(x) = \phi$. We model the distribution parameters $\phi$ as a vector for every image pixel, encoding the likelihood of different labels at that pixel location across protocols.

We model $f_{\theta_f}(x) = \phi$ as a neural network with a UNet architecture, which takes in an image and outputs the parameters $\phi$. Below, we describe the mechanism for sampling segmentation maps from the distribution $p(y_m; \phi|x)$. We then define a loss that encourages segmentation maps for a given protocol $m$ to be semantically consistent across a set of $S$ images from the same biomedical domain.

**Protocol sampling.** We design a new mechanism to produce diverse segmentation maps across different protocols $y_m \sim p(y_m; \phi|x)$. Let a random integer $r_m = (M, K) \sim \mathcal{U}(1, M^{\max}) \times \mathcal{U}(1, K^{\max})$, where $\mathcal{U}$ is the discrete uniform distribution, $M^{\max}$ is a maximum number of label maps and $K^{\max}$ a maximum number of labels for per map. We compute label map $h_{\theta_h}(\phi, r_m) = \hat{y}_m$ given a deterministic function $h_{\theta_h}$. The function first forms an intermediate representation $v_m = e(r_m)$, building on concepts from positional embedding. We concatenate this representation with the distribution parameters at each image location, and use a shallow fully-convolutional network to yield the final segmentation maps:

$$\hat{y}_m = h_{\theta_h}(\phi || v_m) \tag{1}$$
$$= h_{\theta_h}(f_{\theta_f}(x) || e(r_m)). \tag{2}$$

We predict the distributional parameters $\phi$ once, and then efficiently compute $h_{\theta_h}(\phi || e(\{r_m\})) = \{\hat{y}_m\}$ for a set of random integers $\{r_m\}$.

The intermediate representations $v_m = \{v_{m,k}\}$ capture all labels $k$ in protocol $m$. Let $u_m$ and $u_k$ be vector representations corresponding to protocol $m$ and label $k$, inspired by position embedding [113]. We model $v_{m,k} = u_m || u_k$, where $||$ denotes the concatenation operation. Specifically, given integer values $t$, we form vector representation $u_t$ as:

$$u_t^{2j} = \sin(z_{t,2j} + \frac{\pi}{2}), \qquad u_t^{2j+1} = \sin(z_{t,2j} - \frac{\pi}{2}) \tag{3}$$

with

$$z_{t,j} = \frac{t\,\pi}{T} 2^{\frac{2j\,\pi}{J}}, \tag{4}$$

where $u_t^j$ denotes entry $j$ in vector $u_t$, $j = 1, \dots, J$ and $J$ is a hyperparameter determining the size of the vector. We use this formulation to form vectors for any desired $u_m$ and $u_k$, which, in turn, enables us to form $v_m$ for any given protocol. Specifically, the periods $T$ for both vector representations $u_m$ and $u_k$ are determined by the random variable $r_m$, i.e. the values of M and K sampled.

**Inference.** During inference, Pancakes is given a set of images $\{x_s\}$ as input. For each image in the set, Pancakes produces a variable number of label maps $\hat{y}_{s,m,k}$ from various protocols with a variable number of labels $k$ semantically consistent across the images.

### 3.1 Training strategy

Our goal is to enable off-the-shelf multi-protocol segmentation of *any* biomedical image $x$, especially for those not seen during training. To achieve this, we train a single Pancakes model on a wide

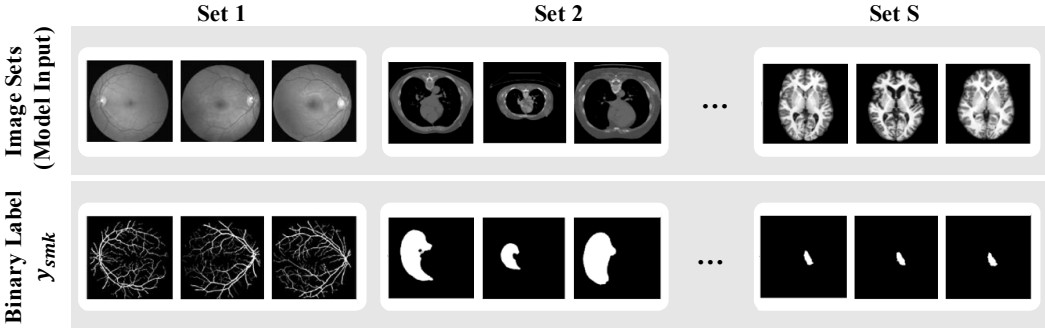

Figure 4: **Example input and binary label available at training.** Images from the same set come from the same domain.

array of biomedical datasets spanning diverse domains and segmentation protocols. In most realistic scenarios, only a subset of the labels (often one label) in a protocol are available in each dataset, and most often only one protocol. At each training iteration, we first sample a dataset from the biomedical collection. If a dataset contains multiple protocols (which is rarely available in public data), we sample a protocol, and then sample a specific label task $t$ within that protocol. Finally, we randomly sample a *set* of images from that dataset, with that label segmented.

At each iteration, for a set of images $\{x_s\}$ with associated ground-truth binary labels $\{y_s\}$, we sample $M$ protocols containing at most $K$ labels each, and predict label maps $\{\hat{y}_{m,s}\}$ for *all* $M$ protocols.

**Loss function.** We design a loss function that *encourages the label maps of any produced protocol to be consistent across the image set*. The loss function also enables learning from data with only a subset of labels segmented in each protocol, and encourages diversity of predicted candidate protocols. We develop this further in the Supplementary Section B.

We define $d_{m,k}(\{\hat{y}_{s,m,k}\}, \{y_s\}) = \mathbb{E}_s[\mathcal{L}_{Dice}(\hat{y}_{s,m,k}, y_s)]$ as the average Dice score for a protocol $m$ and label $k$ across the samples $s$, where $\hat{y}_{m,s,k}$ is the binary map of label $k$ of prediction $\hat{y}_{m,s}$. Denoting $\mathcal{T}$ as the set of all possible tasks and $\mathcal{S}$ the set of all image sets, we optimize model parameters $\theta_f$ and $\theta_h$ by minimizing the loss function

$$\mathcal{L}(\theta_f, \theta_h; \mathcal{T}) = \mathbb{E}_{\mathcal{T}}\mathbb{E}_{\mathcal{S}}[\mathcal{L}_{seg}(\{\hat{y}_{s,m,k}\}, y_s^t)], \tag{5}$$

$$\text{with} \quad \mathcal{L}_{seg}(\{\hat{y}_{s,m,k}\}, y_s^t) = \min_{m,k} d_{m,k}(\{\hat{y}_{s,m,k}\}, \{y_s^t\}). \tag{6}$$

By only penalizing the segmentation of the best performing predicted protocol $m$ and label $k$, the loss function encourages diverse label map samples $\hat{y}_m$ – *at least one* candidate segmentation map matches the ground-truth binary label [58, 96]. By averaging the loss terms across the set, the loss encourages label $k$ of protocol $m$ to refer to the same region in each image. Figure 4 shows examples of the inputs and their corresponding binary labels used in the loss.

**Augmentations.** We apply standard data augmentations to improve generalization, such as Gaussian noise, blur, contrast changes, affine, and elastic transforms. Building on recent strategies [16], we distinguish between two types of augmentations, *in-task* augmentations and *task* augmentations. The *in-task* augmentations are applied independently to each element in a set and aim to increase the diversity of sets. The *task* augmentations are applied consistently across elements of the same set and aid in increasing the number of protocols available at training. A complete list of augmentations and parameters is provided in the supplemental material in F.3.

**Synthetic Data.** To improve the generalization to new domains, we use synthetic data [13, 16, 43] build on *Anatomix* [27]. First, we create maps by sampling binary segmentations from TotalSegmentator [118]. To simulate images from the same set representing the same biomedical region, we first sample a single label map, which is shared across the elements of one set. We then create label maps by applying affine and elastic transforms to the original label map independently. We assign an intensity to all pixels in a given label, and apply several related augmentations to create a synthetic corresponding image set.

## 3.2 Implementation details

For function $f_{\theta_f}(\cdot)$, we use a UNet-like architecture, with convolutional layers of 32 features followed by *PReLU* activation [39]. The function $h_{\theta_h}(\cdot)$ is a series of convolution layers with skip connections. We use the SoftMax function across the $K$ dimension to obtain multi-label segmentation maps, with non-overlapping labels. This ensures that our protocols are *complete*, assigning a label to each image pixel, instead of *partial*, assigning labels to only specific regions of the image. During training, we sample the maximum number of labels $K$, the number of protocols $M$, and the set size $S$ uniformly from a fixed range: $K \in [5, 40], M \in [5, 15], S \in [2, 5]$. Within the same batch, different sets come from different domains. At inference time, $K$ and $M$ can be chosen by the user, while $S$ is determined by the number of images in the set to segment. We use the AdamW optimizer [74] with a learning rate of 0.0001 [56].

## 4 Experiments

We evaluate Pancakes on a broad battery of biomedical segmentation tasks, with three goals: (1) verify that each *protocol* generated by the model is semantically consistent across images; (2) quantify how well Pancakes matches manual segmentations of a provided protocol; and (3) study how design choices during training and inference affect diversity and accuracy. We pay special attention to datasets and imaging domains unseen during training.

**Evaluation.** The evaluation of a method's ability to produce segmentation maps for different plausible protocols, consistently across subjects, is substantially more challenging than evaluation in standard (fixed protocol) segmentation tasks. In most test datasets, only one protocol is provided and only one label is available from that protocol.

We start by assessing if any of the predicted protocols produce a label that closely matches a ground-truth label. We then capture whether this label is consistent across the set. Specifically, we compute the Dice score [107, 109] between the available ground-truth $y_s^t$ and the labels produced for each label map and each image in the set $\hat{y}_{s,m,k}$. We then compute the average across subjects in the set: $\mathbb{E}_s\{Dice(\hat{y}_{s,m,k}, y_s^t)\}$. Finally, we record the produced label that performs best, identified by a specific map $m$ and label $k$. We call this *Set Dice*:

$$\max_{m,k} \mathbb{E}_s\{Dice(\{\hat{y}_{s,m,k}\}, y_s^t)\}. \tag{7}$$

This metric captures: (1) Consistency: a given label should represent the same region in all images in a set; and (2) Accuracy: for each individual image, how well does the best label match the ground-truth segmentation?

Because our goal is to generate a family of anatomically coherent multi-protocol label maps, no single user-provided mask can serve as a universal "ground-truth". Therefore, overlap metrics such as Dice score offer only a partial assessment. We complement our quantitative evaluation with visualizations that illustrate the range of protocols generated by Pancakes. We assess these examples in two aspects: (1) each protocol should be consistent across subjects; and (2) different protocols should generate different, yet still anatomically plausible, partitions. Inevitably, some protocols will resonate more with readers' expectations than others. Our goal in this visualization is to demonstrate the diversity and semantic coherence Pancakes can achieve.

**Data.** We train Pancakes on a large, diverse collection of biomedical data, and evaluate the multi-protocol segmentations produced on images from held-out datasets. We use Megamedical [16, 96, 123], which covers many biomedical domains [2, 3, 6, 11, 12, 14, 16, 19, 22, 29, 31, 32, 35, 37, 41, 42, 47, 49, 51, 52, 54, 61–67, 69, 71–73, 75, 78–82, 87, 93, 94, 98, 101, 103, 105, 106, 108, 110, 127, 129, 132, 133].This dataset has diverse anatomies such as thoracic organs [75], brain [80], eye [78]; and diverse modalities including XRay [47], CT [75], ultrasound [66] and fundus images [46]. We partition this collection into three subgroups. *Training datasets* are used at training time, including model weight optimization and backpropagation. A complete list of the training datasets can be found Table 4 in the supplemental material. *Development datasets*, ACDC [11], PanDental [2], and SpineWeb [132], are *not* used in training but are to evaluate models during development. *Held-out datasets*. These datasets are only used for final evaluation. They include QUBIQ Prostate dataset [83], SCD [94], WBC [133], BUID [3], LIDC-IDRI [5], DDTI [90], and STARE [46].

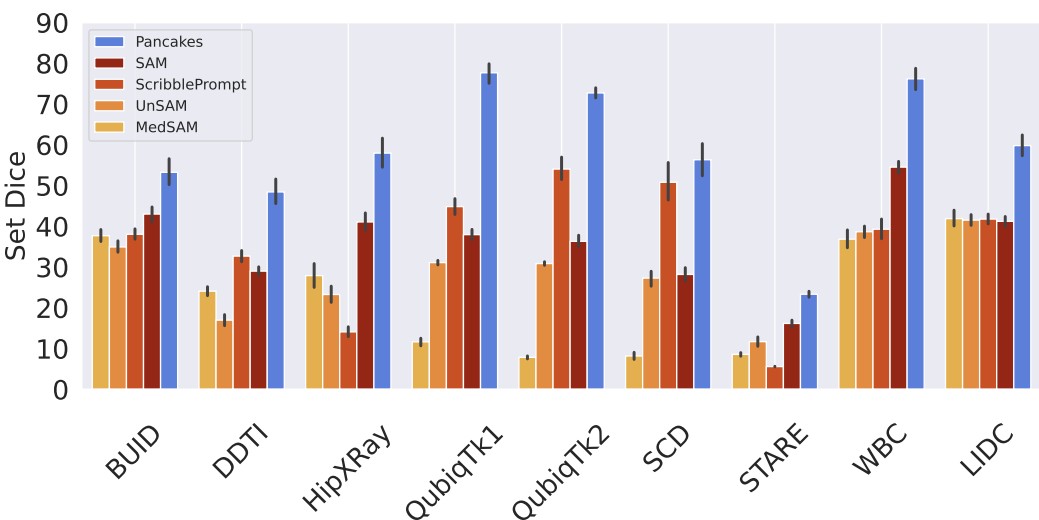

Figure 5: **Pancakes generalizes well to unseen datasets.** We evaluate Pancakes and baselines with the *Set Dice* metric.

The images within each dataset are also split into *training*, *validation*, and *test* splits. We train the models on the *training* split of the *training* datasets. We used the *training* split of the *development* datasets to monitor out-of-distribution capabilities. We report results on the *test* splits of both the *development* (in the supplemental material) and *held-out* datasets. We split the dataset based on subjects, and ensured that there was no train/validation/test subject cross-contamination.

We also synthesized 120, 000 training image-segmentation pairs as described in Section 3. We refer to these examples as *Anatomix* data.

**Benchmarks.** To our knowledge, there are no methods that attempt the same task as Pancakes. We compare our work to four methods that can be used to produce segmentation maps for new images, but none of these were designed to produce *multiple* protocols *consistently* across subjects.

*SAM* [58, 97]: the Segment Anything Model (SAM) is an interactive image segmentation model trained mostly on natural images. SAM involves a whole-image-segmentation mode, which produces a grid of simulated clicks over the whole image, removes high-overlapping regions and selects the most likely masks using a confidence threshold. We use this mode, and produce diverse whole-image segmentations by varying the confidence threshold.

*ScribblePrompt* [123]: ScribblePrompt (SP), is an interactive segmentation tool trained on Megamedical. We use the SP-SAM, which performs best on clicks, and obtain multi-protocol and multi-label using the same whole-image strategy we used for SAM.

*MedSAM* [76]: trained specifically on medical data, this model uses a SAM-like architecture and is optimized for bounding box interactions. We produce whole-image segmentations using the SAM whole-image segmentation mode.

*UnSAM* [116]: trained on a curated set of natural images, this model produces segmentation masks using a DINO [20] backbone (pretrained ResNet50 [40]) and the Mask2Former [21] mask decoder. UnSAM produces a list of labels to serve as label maps. We use the *UnSAM+* model version, trained on a part of the SA-1B dataset [58]. To produce segmentations with diverse protocols and labels, we use the UnSAM's existing whole-image segmentation scheme.

## 5   Results

For our main evaluation, all error bars are 95% confidence interval using 1000 bootstraps of all sets. Additional per-dataset performance and ablations are shown in the supplemental material.

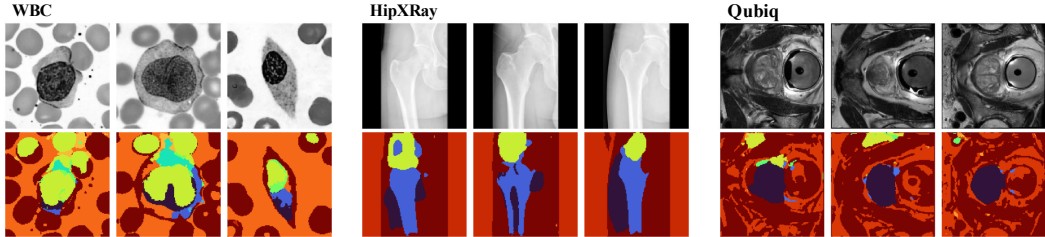

Figure 6: **Pancakes set consistency.** For the same protocol, each label, identified by color, is assigned to similar structures across a set of images.

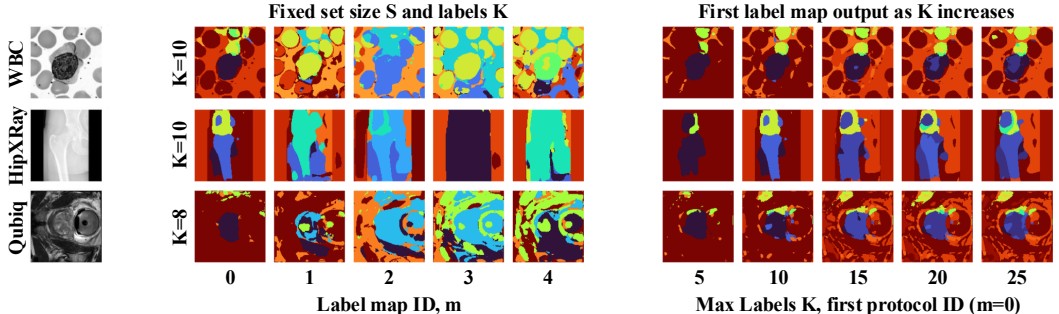

Figure 7: **Effect of M and K.** Left: Input image for the first subject in the set ($s = 0$) with set size $S = 3$. Middle: Pancakes can produce diverse label maps with fixed $K$ labels per protocol. Right: Increasing the maximum number of labels $K$ tends to lead to finer structures in the produced label maps.

Quantitatively, Figure 5 shows that Pancakes outperforms the baselines on all unseen datasets, often by a margin of more than 20 Dice points. For a more detailed assessment, we separately report accuracy, consistency, and the effects of hyperparameters, and visualize results in several scenarios.

**Accuracy versus consistency.** Figure 8 shows that Pancakes performs well on *both* segmentation accuracy and semantic consistency across images. For individual accuracy (set size $S = 1$), which does not penalize semantic consistency across images, Pancakes performs similarly to SAM, and is superior to all the other baselines. As the set size $S$ increases, all baselines fail to produce semantically *consistent* segmentations, while Pancakes segmentations remain consistent across the set, leading

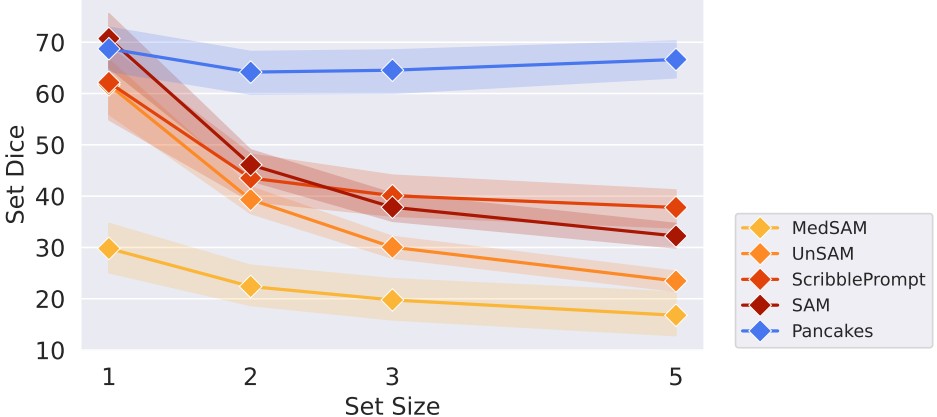

Figure 8: **Pancakes is both consistent and accurate compared to baselines.** When evaluated solely on accuracy ($S = 1$), Pancakes is comparable to SAM and outperforms baselines. As $S$ increases, Pancakes is the only model whose performance is not degraded.

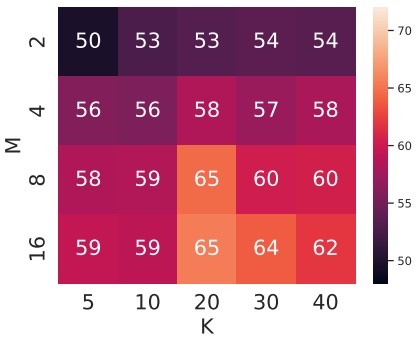

Figure 9: **Influence of segmentation maps** $M$ **and labels** $K$ **on segmentation quality for** $S = 3$.

Table 1: **Inference time and number of parameters.** Pancakes has substantially fewer parameters and is significantly faster than baseline methods.

|  | #Param | S=1 (sec.) | S=3 (sec.) |
|---|---|---|---|
| **Ours** | **0.22 M** | **0.10 ± 0.04** | **0.12 ± 0.03** |
| SAM | 641M | 3.13±0.16 | 2.94 ± 0.24 |
| SP | 93.7M | 1.99 ± 0.13 | 1.85 ± 0.18 |
| MedSAM | 93.7M | 2.12 ± 0.17 | 1.94 ± 0.18 |
| UnSAM | 23M | 0.54 ± 0.012 | 0.45 ± 0.03 |

to a steady *Set Dice*. We hypothesize that SAM is better than biomedical baselines because it was trained on a wider variety of labels and images. Therefore, SAM is less prone to task-overfitting. In the Supplementary Section C, we also report results for various M and K, and also evaluate using the *Set Surface Distance* and *Set IoU* metrics.

Figure 6 illustrates Pancakes predictions for a fixed protocol, and highlights visually the semantic consistency across the images of a set. Additional visualizations with the baselines are shown in Section C.1.

Overall, Pancakes outperforms or matches the other methods in producing plausible protocols, and outperforms all methods by a substantial margin in producing semantically consistent protocols.

**Influence of M and K.** Figure 7 illustrates the diversity of protocols captured in Pancakes segmentation outputs. They are produced by fixing the set size $S$ and the maximum number of labels $K$ and performing one forward pass through the network. The label maps are different from one another across protocol ID $m$ and maximum labels K, capturing structures at various granularity levels. Figure 7 also shows that increasing the number of labels $K$ per protocol results in segmentations of finer structures.

Figure 9 captures the quantitative effects of the number of label maps $M$ and maximum label number $K$ for a fixed set size $S$. Producing more protocols leads to better performance in general, while performance as a function of $K$ is more variable. We hypothesize that this arises from the non-overlapping nature of labels within each protocol. Ambiguous regions that could belong to multiple labels require separate protocols to represent each plausible interpretation, so having more protocols enables the model to better capture such ambiguities.

**Efficiency.** We study runtime requirements by running all the models on the HipXRay [37] test split. Table 1 shows the average run time across 1000 trials. Pancakes is substantially faster than all baselines, because of its efficient fully-convolutional architecture, leading to a leaner model and fewer parameters.

**Analysis.** We use *Set Dice* with $S = 3$, and use $M = 8$ and $K = 20$ for Pancakes. As supplementary Figure 15 shows, performance on the development datasets saturates at $M = 8$ and $K = 20$, with only marginal gains achieved by further raising $M$. We therefore chose to use these parameters as a balance between performance and the number of structures clinical users might actually expect in practice.

**Influence of synthetic data.** We compare versions of Pancakes trained with real data only, synthetic data only, or both. When trained with both real and synthetic data, Pancakes consistently outperforms the other variants in Dice score ($p < 0.05$ with a paired Student-t test), for $M = 16$, as shown in Table 2. We find that for $M = 8$, the performance change is not statistically significant.

**Pancakes and interactive segmentation.** If a user has a particular segmentation protocol in mind, but there are no existing tools for it, they can choose the protocol from Pancakes' outputs that best

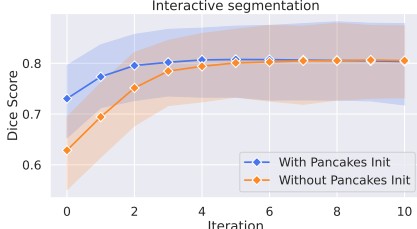

Figure 10: **Interactive Segmentation with Pancakes initialization.** Pancakes can be used as an initialization for interactive segmentation to obtain high quality labels faster.

Table 2: **Influence of synthetic data across set size.** For $M = 16$, training with both synthetic data and Megamedical yields a small yet consistent improvement over only training with Megamedical.

|  | Both | Megamedical | Synthetic |
|---|---|---|---|
| S = 1 | **73.2 ± 5.5** | 71.1 ± 6.2 | 56.3 ± 4.9 |
| S = 2 | **67.3 ± 6.0** | 65.8 ± 6.7 | 45.8 ± 5.0 |
| S = 3 | **67.4 ± 6.0** | 65.7 ± 6.8 | 44.3 ± 5.4 |
| S = 5 | **68.4 ± 5.6** | 67.4 ± 6.9 | 42.7 ± 5.5 |

aligns with their intended use. This choice will often suffice. In cases where Pancakes' segmentations are not sufficiently accurate–for example, when targets are far from the training distribution–the segmentation maps can offer excellent initializations to interactive segmentation systems such as ScribblePrompt [123]. To demonstrate this, we compared using ScribblePrompt with and without initialization using Pancakes' predictions: (1) on average, Pancakes' predictions can be improved by 5 Dice points with a *single interactive click*, for set sizes larger than one. (2) on average, using ScribblePrompt with a Pancakes-initialized segmentation reduces the number of required interactions by half. If used alone, it takes ScribblePrompt 5 to 8 clicks for the prediction quality to plateau, while, when initialized with Pancakes' predictions, ScribblePrompt can reach the same results in 3 to 4 clicks.

# 6 Assumptions & Social Impact

In this work, we made several core assumptions. First, we assume that the user has an idea of the desired labels and can select a few $K$ values *a priori*. Pancakes is intended for biomedical experts. We assume that as they use the tool, they can identify the mapping between a label and a known structure. Second, we assume that visualizing several protocols simultaneously is reasonable. Third, we assume that while we trained and evaluated on a diverse set of medical images, we likely did not capture all biomedical image types and domains that a user might encounter. Fourth, Pancakes is not intended to replace existing clinically validated segmentation protocols. If there is an existing tool for a particular protocol, we would advise using it.

We aim for this work to inspire a new approach to using foundation models in biomedical imaging. Pancakes is designed to be efficient and accessible, especially in resource-constrained settings. It can support both data annotation and exploratory analysis, as well as serve as a tool for developing future foundation models. However, this version is intended for research use only. While trained on a broad collection of small biomedical datasets, we have not evaluated it for potential societal biases.

# 7 Conclusion

We introduced Pancakes, a new framework that predicts segmentation maps for multiple protocols in previously unseen biomedical imaging domains, with each protocol being semantically consistent across images. Pancakes estimates a distribution over plausible label map protocols and provides a mechanism to sample multiple segmentation maps from this distribution.

Predicting plausible segmentations in a new domain, without prior knowledge of the number of labels or their associated shapes, is a challenging task. Our experiments demonstrate that Pancakes achieves state-of-the-art performance on seven held-out datasets. We believe this work addresses an important, previously unaddressed problem, enabling biomedical researchers to segment entire collections of images without requiring manual annotations or example segmentations. This can, in turn, substantially speed up downstream biomedical studies.

# 8   Acknowledgement

We would like to thank Neel Dey for the very helpful discussions and feedback. This research was supported by the National Institute of Biomedical Imaging and Bioengineering of the National Institutes of Health under award number R01EB033773, the Eric and Wendy Schmidt Center at the Broad Institute of MIT and Harvard, Quanta Computer Inc. Some of the computation resources required for this research was performed on computational hardware generously provided by the Massachusetts Life Sciences Center.

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

# A  Table of Content

We divide this supplemental material into five major parts:

**Frequently asked questions ( B).** We address some questions that have been asked by colleagues about this work. We think this might be relevant to reviewers if this remains unclear in the paper.

**Additional analysis on *Held-out* Datasets ( C).** We report results for the *Set Surface Distance* and *Set IoU* metrics. We analyze further architectural choices and their impact on performance. This includes the size of the set and the choice of $M$ and $K$. We provide per-dataset performance when results are shown aggregated in the main paper.

**Additional analysis on *Development* data ( D).** We present performance for the *Development* datasets, which were used to evaluate the generalization capabilities of the model.

**Training infrastructure ( E).** We detail our training infrastructure and requirements.

**Data ( F).** We provide additional information on Megamedical and Anatomix as well as the augmentations applied.

# B  Frequently Asked Questions

**Instead of Pancakes, could a user fine-tune a model on a set of curated images?** Pancakes tackles the prevalent scenario where there is no preexisting annotated data. It relieves the user of the requirement imposed by existing methods: annotating sufficient images to use for few-shot or in-context methods.

**What is the difference between Pancakes and stochastic segmentation methods [10, 59]?** Pancakes is focused on segmenting images in **new** tasks, not seen at training, while existing methods are specifically designed to model stochasticity in a given pre-specified task, essentially tackling a different problem. While Pancakes can also capture diversity for a fixed *single* protocol like existing models do, we model the substantially more challenging problem of presenting *different* plausible (multi-label) protocols for the new task.

**What does it mean for segmentations to be consistent across a set?** If a label ID appears across a set of anatomically similar images, this ID semantically corresponds to the same ROI in each image. For example, if label 1 represents the hippocampus in the first image, it should also represent the hippocampus in the second image.

**Why is consistency important?** Consistency is critical when the user wants to analyze the same set of structures for multiple images as is prevalent in population and longitudinal studies.

**Is it possible to achieve consistency through post-processing if a method is inconsistent?** For previous methods, matching segmentation maps across scans is not trivial for several reasons. Among them: (1) The segmentation maps in different scans are often so different that there are no clear labels to match. The same segmentation model could segment images on different bases and different images can contain different structures (2) Even when the same anatomical structure is segmented in two different images, they usually don't share the same label ID or don't have the same size as shown in Figure 2. This makes automated matching as post-processing step ill-posed.

**How are the sets treated at training?** We train with input tensors of dimension $B \times S \times C \times H \times W$, for batch $B$, set $S$, channel $C$, image height $H$ and width $W$. We flatten these tensors to $(B \times S) \times C \times H \times W$ so that we can use architectures based on 2D-convolutions and 2D-UNets.

**What is the intuition behind the min Dice loss?** Using a regular expectation over the candidate label maps would lead the model to regress to the mean. This might be particularly harmful when there is high uncertainty around the set of plausible labels to output. Instead, using the best values leads to more diverse predictions. We refer to [96] for a more extensive comment on this type of loss.

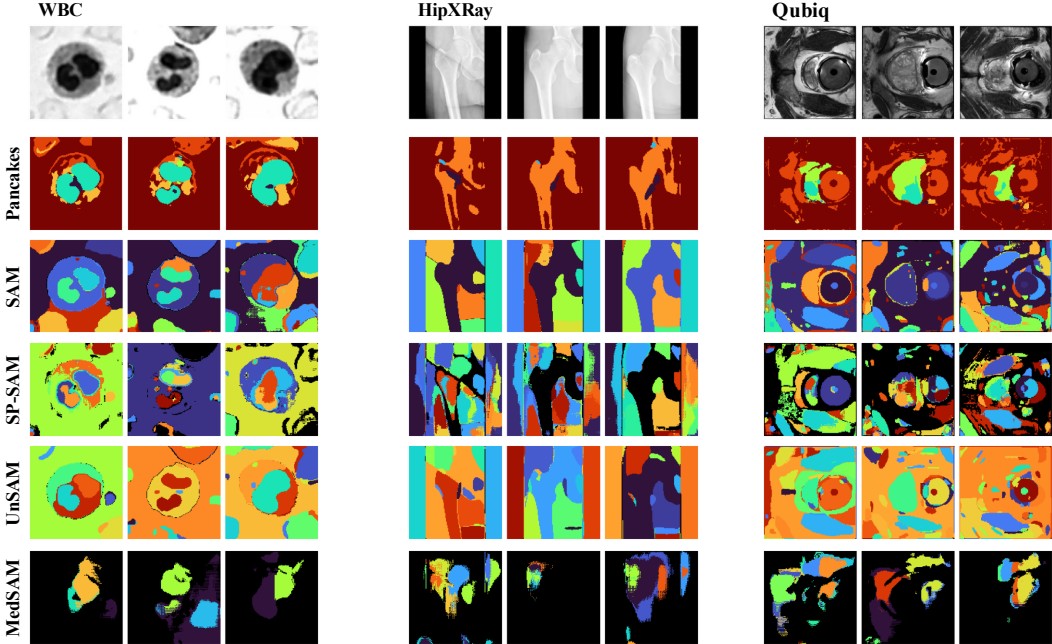

Figure 11: **Pancakes and baselines set consistency.** Evaluated on 3 held-out datasets (left to right: WBC, HipXRAY, QUBIQ Prostate), Pancakes is the method that is the most consistent across predictions compared to the baselines.

## C  Additional Analysis on *Held-out* Datasets

We present more in-depth results for the *held-out* datasets that remained unseen until the final phase of evaluation.

### C.1  Qualitative comparison with baselines

We visually compare predictions for three held out datasets: WBC, HipXRay and QubiqProstate. Figure 11 shows that even though some baselines produce more accurate individual labels, Pancakes provides the most consistent labels across images of the same set.

### C.2  Interpolation

This additional visualization explores the space of protocols $M$ for multiple numbers of labels: $K$. For $M = 16$, we sample the 16 protocols available for different $K$ values: $[5, 10, 15, 20, 25]$. We show in Figure 12 how this impacts the produced label maps for two samples in the WBC dataset. We find that segmentation maps that come from protocols *close* to one another in the embedding space are similar. Interestingly, the label space interpolated is relatively smooth, as indicated by the color and shape variations.

### C.3  Additional Metrics

In Figure 13, we report results evaluated using two additional metrics: *Set IoU* and *Set surface distance*. They are computed in the same way as *Set Dice* but replacing Dice score with IoU and Surface Distance respectively. For the surface distance, we keep the percentile 95. When the set size is one ($S = 1$), the set metric reduces to the standard metric, ignoring consistency.

### C.4  Per set size performance

We report additional per-set-size results that were summarized in the main paper.

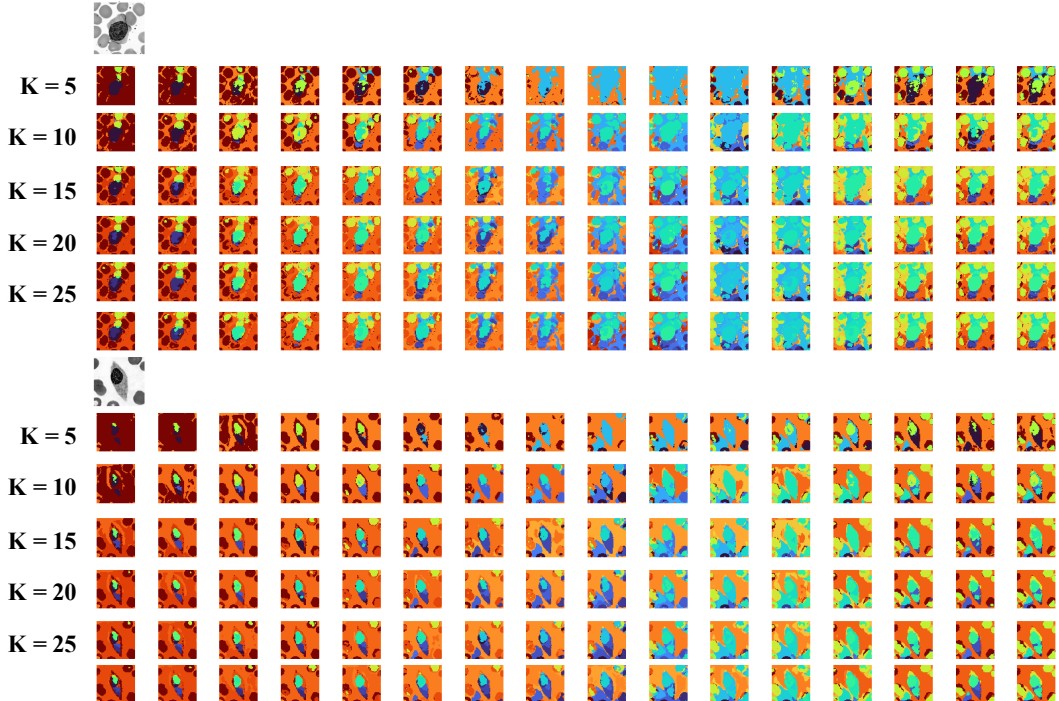

Figure 12: **Interpolation Analysis.** By covering the space of protocol (rows) for several fixed $K$ values, we observe that label maps resulting from protocols *close* to another in the embedding space are similar.

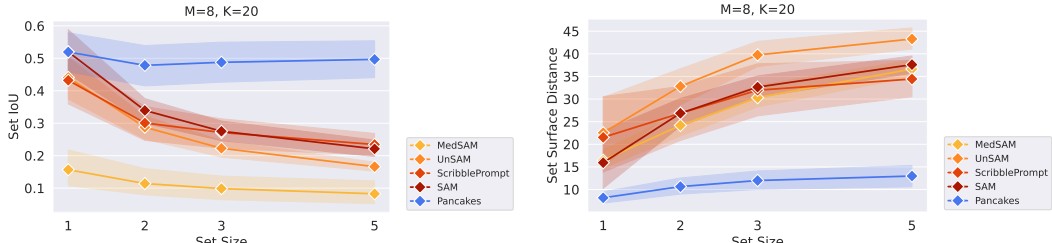

Figure 13: **Evaluation on additional metrics: IoU and surface distance.** For individual predictions ($S = 1$), Pancakes is comparable to SAM and outperforms baselines. As the set size $S$ increases, Pancakes is the only model whose performance is not severely degraded, as it can produce consistent segmentations.

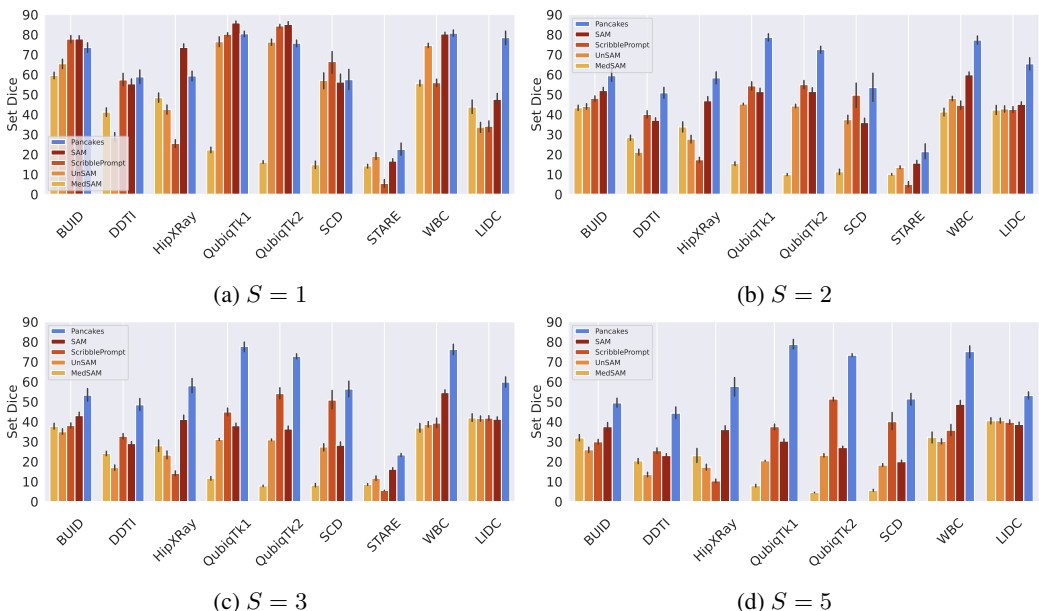

(a) $S = 1$           (b) $S = 2$

(c) $S = 3$           (d) $S = 5$

Figure 14: **Per-dataset *Set Dice* performance under various set sizes.** We evaluate Pancakes and baselines with various set sizes: 1, 2, 3 and 5. For Pancakes, we evaluate with $M = 8$ and $K = 20$.

**Per-dataset *Set Dice* performance.** This experiment evaluates how segmentation accuracy and label consistency vary jointly with set size. Figure 14 shows that Pancakes's improvement over the baselines increases with the set size. Figure 14a focuses on accuracy only as the set size is one ($S = 1$). In this case, Pancakes is comparable to the baselines (Figure 14a). We observe a large variability across different datasets.

**Influence of $M$ and $K$.** Figure 15 shows the influence of the number of protocols and labels on the Pancakes predictions as the set size varies. As the number of protocols increases, performance increases. Best performing $K$ seems to be in the middle of the range for $K$=20. Results are consistent across set sizes.

# D    Additional Analysis on *Development* Datasets

We include performance on the *Development* datasets that were used to evaluate generalization capabilities during model development: ACDC [11], PanDental [2] and SpineWeb [132].

We evaluate on the test split of all datasets. When reporting performance with a certain set size, we exclude tasks with subjects fewer than this set size. (For example, the test split of SpineWeb has only 2 subjects, so we do not include it in set size 3 and 5 experiments.) We report Pancakes performance with $M = 8$ and $K = 20$. Figure 16 shows Pancakes demonstrates both accurate and consistent predictions compared to the baselines.

**Per-Dataset *Set Dice* Performance for Different Set Sizes.** Figure 17 shows per-dataset performance, separated by set size.

**Influence of $M$ and $K$.** Figure 18 shows effect of number of protocols and labels per protocol on prediction accuracy, as set size varies. The trend aligns with results using the held-out datasets (C.4).

**Learning a complete protocol.** The primary objective of Pancakes is to propose a diverse set of consistent protocols for an image group. We do not enforce constraints across protocols produced, but at times it would be good to do so. For example, symmetric labels (left and right ventricles or posterior and anterior hippocampus) in the same protocol may be desirable. One way to incorporate this into the Pancakes framework is to apply the loss function to two randomly chosen segmentation

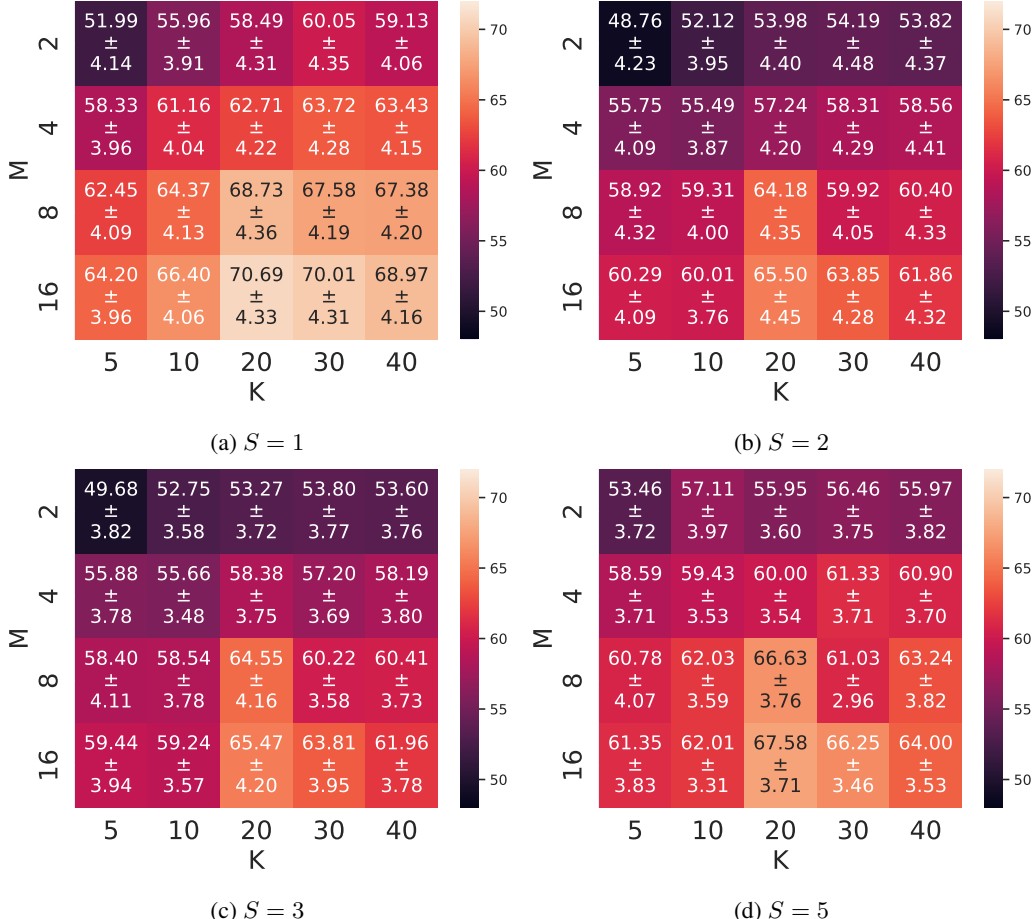

Figure 15: **Influence of varying $M$ and $K$ for various set sizes on the *Held-out* datasets.** We evaluate Pancakes with various set sizes, number of protocols ($M$) and number of labels in each protocol ($K$).

targets in a multi-label dataset, rather than one. As a preliminary experiment, we ran an experiment on the OASIS brain dataset, and found it effective at learning a protocol as shown in Figure 19. We are planning to explore this more in future work.

# E  Training Infrastructure

Our model was trained using 45G of memory on a single node of an NVIDIA DGX A100 machine using two cores. We use a batch size of 1 and the AdamW optimizer with a learning rate of 0.0001. We use PReLU activations and convolution layers with 32 features, kernel size 3 and stride 1.

# F  Data

## F.1  Megamedical

We train our main experiment on Megamedical [16, 96, 123]. The images are 2D and resized to $128 \times 128$. Complete tables of the datasets, split by data subgroup (*Training*, *Development*, *Held-out*) are shown in Tables 4, 5, and 6. Megamedical covers a wide range of modalities (MRI, ultra-sound, XRay) and anatomies (organs, bones, substructures and fine structures like vessels).

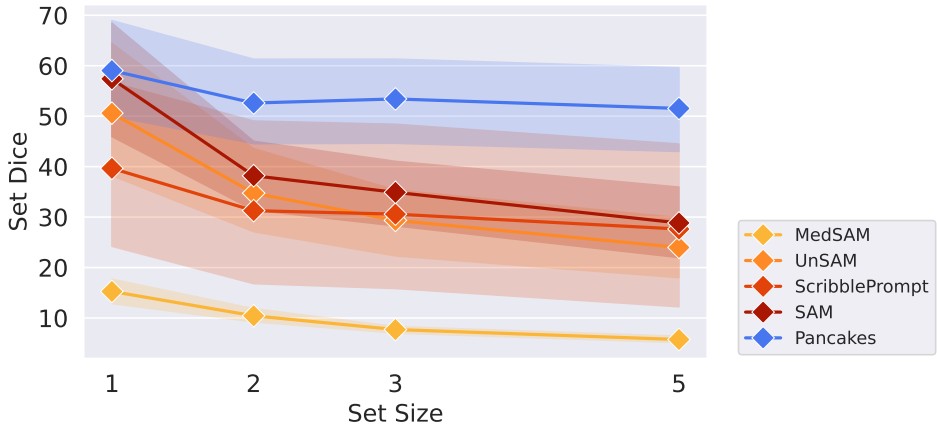

Figure 16: **Pancakes is both consistent and accurate compared to baselines.** We evaluate Pancakes and the baselines on *Development* datasets: ACDC, PanDental and SpineWeb. When evaluated solely on accuracy ($S = 1$), Pancakes is comparable to SAM and outperforms the other baselines. As $S$ increases, Pancakes is the only model whose performance is not degraded.

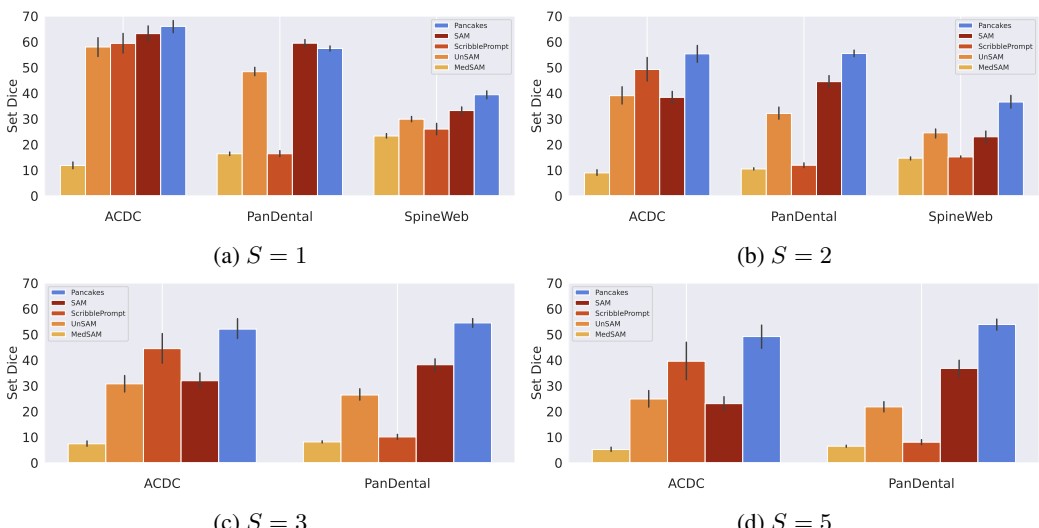

Figure 17: **Per-dataset *Set Dice* performance under various set sizes for the *Development* datasets.** We evaluate Pancakes and baselines on development datasets (ACDC, PanDental and SpineWeb) with various set sizes: 1, 2, 3 and 5. If a dataset has fewer subjects than the set size, we exclude it. For Pancakes, we use $M = 8$ and $K = 20$.

## F.2 Anatomix

Building on [13, 16, 43] and specifically on Anatomix [27], we use synthetic data to complement Megamedical and limit overfitting. To generate Anatomix label maps, we sample a random set of 3D labels from the TotalSegmentator dataset [118]. We sample between 20 and 40 labels and generate a $128 \times 128 \times 128$ label map. Once this 3D label map is generated, we randomly sample an axis and then a slice between slice ID 25 and 100. With probability 50%, we will *split* labels. In that case, labels are reassigned so that a given label can only be composed of contiguous pixels. We also assign to background (label ID 0) any label whose size is smaller than 20 pixels. This gives us the label map template for a given set, from which we are going to generate the images and label maps for each element in the set. We generate the images by randomly assigning an intensity value to each label. We then apply a series of augmentations that are independent for each element in the set. These augmentations include: Gaussian blur, Gaussian noise, Perlin noise, elastic and affine transforms,

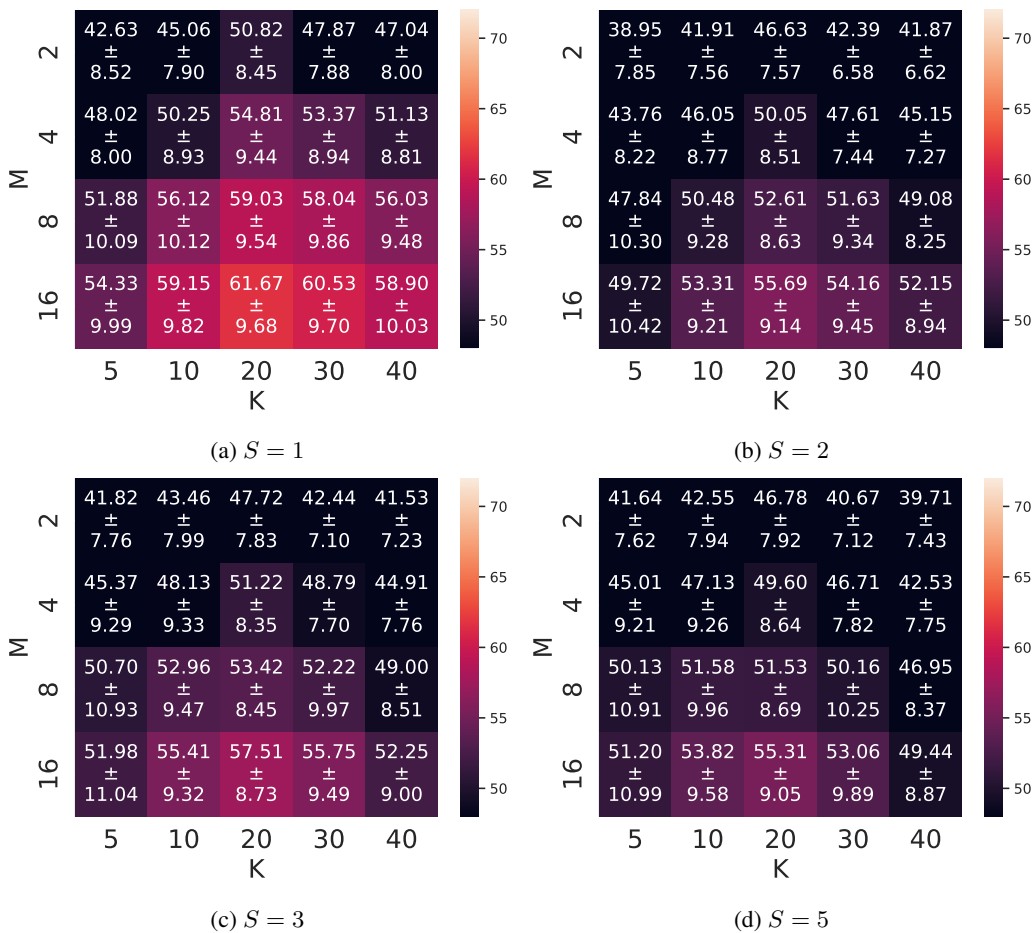

Figure 18: **Influence of varying $M$ and $K$ in various set sizes for the *Development* datasets.** On development datasets, we evaluate Pancakes with various set sizes, number of protocols ($M$) and number of labels in each protocol ($K$).

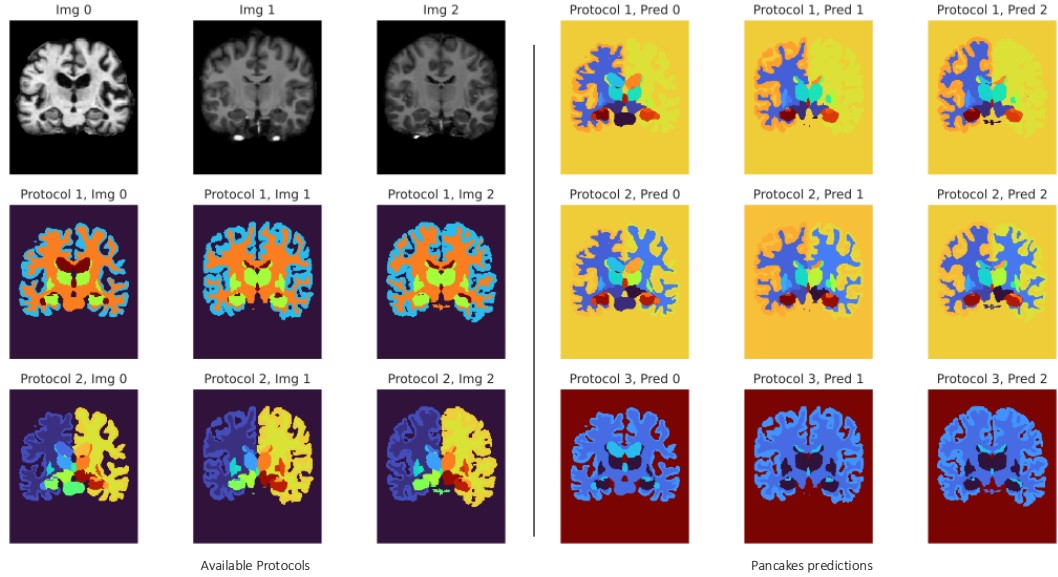

Figure 19: **Learning a full protocol.** Training on OASIS, we learn accurately a complete protocol with our loss modification.

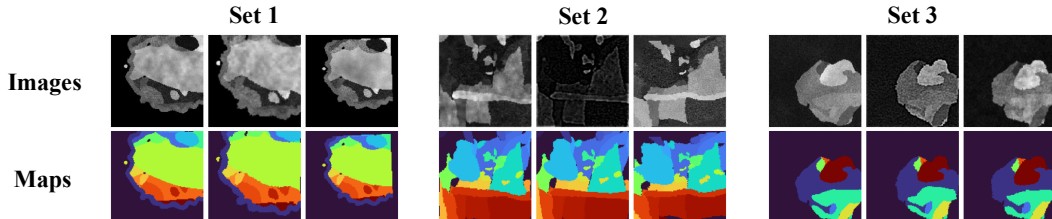

Figure 20: **Anatomix Examples.** Label maps and images within a set are very similar to one another, sharing similar relative locations for each structure. The main difference lies in the augmentations applied, to the images individually (Gaussian noise for example), but also to both the images and label maps (elastic deformation for example).

and contrast variations. To sample a binary ground truth label at training, we sample a label that is present in each image of the set. Example sets generated are shown in Figure 20.

## F.3   Data Augmentation

At training, we apply a series of augmentations to our set to improve generalization capabilities. We distinguish between two types of augmentations, *within-set* augmentation — applied independently to each element in the set — and *across-set* augmentation — applied consistently to each element in the set. For each augmentation sampled, the corresponding parameters are sampled uniformly from a pre-defined range. Table 3 shows the list of all augmentations applied, the probability of each augmentation per iteration and the parameter ranges we sample from.

Table 3: **Augmentations used to train** We apply augmentations either independently within a set (Top) or to all the elements of a set (Bottom). We sample each parameter from the uniform distribution ($\mathcal{U}$) within the ranges defined in the *Parameters* column.

(a) Within-Set Augmentation

| **Augmentations** | $p$ | Parameters |
|---|---|---|
| Random Affine | 0.25 | degrees $\sim \mathcal{U}(-25, 25)$ 
 translate $\sim \mathcal{U}(0, 0.1)$ 
 scale $\sim \mathcal{U}(0.9, 1.1)$ |
| Brightness Contrast | 0.5 | brightness $\sim \mathcal{U}(-0.1, 0.1)$, 
 contrast $\sim \mathcal{U}(0.5, 1.5)$ |
| Elastic Transform | 0.8 | $\alpha \sim \mathcal{U}(1, 2.5)$ 
 $\sigma \sim \mathcal{U}(7, 9)$ |
| Sharpness | 0.25 | sharpness$= 3$ |
| Flip Intensities | 0.5 | None |
| Gaussian Blur | 0.25 | $\sigma \sim \mathcal{U}(0.1, 1)$ 
 k=5 |
| Gaussian Noise | 0.25 | $\mu \sim \mathcal{U}(0, 0.05)$ 
 $\sigma \sim \mathcal{U}(0, 0.05)$ |

(b) Across-Set Augmentation

| **Augmentations** | $p$ | Parameters |
|---|---|---|
| Random Affine | 0.5 | degrees $\sim \mathcal{U}(0, 360)$ 
 translate $\sim \mathcal{U}(0, 0.2)$ 
 scale $\sim \mathcal{U}(0.8, 1.1)$ |
| Brightness Contrast | 0.5 | brightness $\sim \mathcal{U}(-0.1, 0.1)$, 
 contrast $\sim \mathcal{U}(0.8, 1.2)$ |
| Gaussian Blur | 0.5 | $\sigma \sim \mathcal{U}(0.1, 1.1)$ 
 $k = 5$ |
| Gaussian Noise | 0.5 | $\mu \sim \mathcal{U}(0, 0.05)$ 
 $\sigma \sim \mathcal{U}(0, 0.05)$ |
| Elastic Transform | 0.5 | $\alpha \sim \mathcal{U}(1, 2)$ 
 $\sigma \sim \mathcal{U}(6, 8)$ |
| Sharpness | 0.5 | sharpness$= 5$ |
| Horizontal Flip | 0.5 | None |
| Vertical Flip | 0.5 | None |

Table 4: **Collection of datasets in Megamedical used for training.** The entry # of scans is the number of unique (subject, modality) pairs for each dataset.

| Dataset Name | Description | # of Scans | Image Modalities |
|---|---|---|---|
| AMOS [49] | Abdominal organ segmentation | 240 | CT, MRI |
| BBBC003 [72] | Mouse embryos | 15 | Microscopy |
| BBBC038 [17] | Nuclei images | 670 | Microscopy |
| BrainDev. [33, 34, 63, 104] | Adult and Neonatal Brain Atlases | 53 | multi-modal MRI |
| BRATS [6, 7, 84] | Brain tumors | 6,096 | multi-modal MRI |
| BTCV [65] | Abdominal Organs | 30 | CT |
| BUS [128] | Breast tumor | 163 | Ultrasound |
| CAMUS [66] | Four-chamber and Apical two-chamber heart | 500 | Ultrasound |
| CDemris [51] | Human Left Atrial Wall | 60 | CMR |
| CHAOS [53, 55] | Abdominal organs (liver, kidneys, spleen) | 40 | CT, T2-weighted MRI |
| CheXplanation [101] | Chest X-Ray observations | 170 | X-Ray |
| CT-ORG[98] | Abdominal organ segmentation (overlap with LiTS) | 140 | CT |
| DRIVE [110] | Blood vessels in retinal images | 20 | Optical camera |
| EOphtha [25] | Eye Microaneurysms and Diabetic Retinopathy | 102 | Optical camera |
| FeTA [89] | Fetal brain structures | 80 | Fetal MRI |
| FetoPlac [9] | Placenta vessel | 6 | Fetoscopic optical camera |
| HMC-QU [26, 57] | 4-chamber (A4C) and apical 2-chamber (A2C) left wall | 292 | Ultrasound |
| I2CVB [67] | Prostate (peripheral zone, central gland) | 19 | T2-weighted MRI |
| IDRID [93] | Diabetic Retinopathy | 54 | Optical camera |
| ISLES [42] | Ischemic stroke lesion | 180 | multi-modal MRI |
| KiTS [41] | Kidney and kidney tumor | 210 | CT |
| LGGFlair [15, 82] | TCIA lower-grade glioma brain tumor | 110 | MRI |
| LiTS [12] | Liver Tumor | 131 | CT |
| LUNA [105] | Lungs | 888 | CT |
| MCIC [32] | Multi-site Brain regions of Schizophrenic patients | 390 | T1-weighted MRI |
| MSD [106] | Collection of 10 Medical Segmentation Datasets | 3,225 | CT, multi-modal MRI |
| NCI-ISBI [14] | Prostate | 30 | T2-weighted MRI |
| OASIS [45, 80] | Brain anatomy | 414 | T1-weighted MRI |
| OCTA500 [69] | Retinal vascular | 500 | OCT/OCTA |
| PAXRay [103] | Thoracic organs | 880 | X-Ray |
| PROMISE12 [71] | Prostate | 37 | T2-weighted MRI |
| PPMI [81] | Brain regions of Parkinson patients | 1,130 | T1-weighted MRI |
| QUBIQ [83] | Brain, kidney, pancreas | 209 | MRI T1, Multimodal MRI, CT |
| ROSE [78] | Retinal vessel | 117 | OCT/OCTA |
| SegTHOR [64] | Thoracic organs (heart, trachea, esophagus) | 40 | CT |
| ToothSeg [47] | Individual teeth | 598 | X-Ray |
| WMH [62] | White matter hyper-intensities | 60 | multi-modal MRI |
| WORD [75] | Organ segmentation | 120 | CT |

Table 5: **Development datasets**. Datasets used to evaluate the generalization capabilities of our model and model development.

| Dataset Name | Description | # of Scans | Image Modalities |
|---|---|---|---|
| ACDC [11] | Left and right ventricular endocardium | 99 | cine-MRI |
| PanDental [2] | Mandible and Teeth | 215 | X-Ray |
| SpineWeb [132] | Vertebrae | 15 | T2-weighted MRI |

Table 6: **Held-out datasets**. Datasets that remained unseen until the final evaluation phase.

| Dataset Name | Description | # of Scans | Image Modalities |
|---|---|---|---|
| BUID [3] | Breast tumors | 647 | Ultrasound |
| DDTI [91] | Thyroid | 472 | Ultrasound |
| HipXRay [37] | Ilium and femur | 140 | X-Ray |
| QUBIQ [83] | Prostate | 209 | MRI T1, Multimodal MRI, CT |
| SCD [94] | Sunnybrook Cardiac Multi-Dataset Collection | 100 | cine-MRI |
| LIDC-IDRI [5] | Lung Nodules | 1,018 | CT |
| STARE [46] | Blood vessels in retinal images | 20 | Optical camera |
| WBC [133] | White blood cell and nucleus | 400 | Microscopy |

