# OpenReview forum: "Pancakes: Consistent Multi-Protocol Image Segmentation Across Biomedical Domains"
_NeurIPS.cc/2025/Conference — NeurIPS 2025 poster_

### Official Review · Reviewer_jNMX · 2025-06-28

**Clarity:** 2
**Significance:** 2
**Originality:** 2
**Rating:** 4
**Confidence:** 5

**Summary:**

The authors proposed a method that can segment biomedical image with different protocols, named Pancakes. It is framework that can automatically generate multi-label segmentation maps for multiple plausible protocols given a new image from a previously unseen domain. Although the idea is interesting, there are some major concerns about the validity of the measurement, and if this is really clinically meaningful.

**Questions:**

1. One of the most important concepts of biomedical image segmentation is to accurately segment the targeted tissue/organ/region that have been previously clinically well defined. Only in this way, the segmentation map can be useful for clinical downstream task, like evaluation fat ratio, organ size, cancer aggressiveness, etc. Although the Pancakes can provide multiple different segmentation maps from potential protocols, then who should confirm the clinical meaning of these protocols. And also, how we can utilize the segmentation maps in the downstream tasks? For example, you get a segmentation map, and it looks like for fat segmentation, can you CONFIDENTLY STATE this is indeed for fat segmentation, but not mis-segmented muscle segmentation? If you can, what is your reasoning behind? If you cannot, then I would say this is a relatively big problem.
2. “The images within each dataset are also split into training, validation, and test splits.” Did you split the dataset based on images, or based on patients?
3. “UNet-like architecture, with convolutional layers of 32 features followed by PReLU activation”, surprisingly, the original UNet can perform better than other baseline approaches. Like in Table 1, the comparisons are very eye-catching, and a bit not that realistic to me. I would like to ask the authors to dig into why this can happen and provide corresponding clues.
4. “We start by assessing if any of the predicted protocols produce a label that closely matches a ground truth label.” It is “closely matches” defined by Dice score enough? Is this assessment clinically verified by the corresponding specialists? For example, vessels, they are more caring about connectivity/topology, etc.
5. “Finally, we only record the label that performs best, identified by a specific map m and label k. We call this Set Dice” This is the problem I mentioned in the first bullet point. How can you use this outcome in downstream clinical task is a really big problem.

**Ethical Concerns:**

["NO or VERY MINOR ethics concerns only"]

**Final Justification:**

The authors resolved my comments well, and thus rating raised.

**Quality:**

2

**Strengths And Weaknesses:**

Strengths:
1. The authors noticed an interesting question in biomedical image segmentation domain, and proposed a framework trying to resolve it.
2. Relatively enough number of medical imaging and organ/tissue were evaluated.

Weaknesses:
1. The authors seems to ignore the clinical meaning of the segmentation masks, which makes the work a bit not practical. One of the most important concepts of biomedical image segmentation is to accurately segment the targeted tissue/organ/region that have been previously clinically well defined. Only in this way, the segmentation map can be useful for clinical downstream task, like evaluation fat ratio, organ size, cancer aggressiveness, etc. Although the Pancakes can provide multiple different segmentation maps from potential protocols, no one can confirm its clinical usefulness.
2. The author proposed a new measurement “Set Dice”. It is kind of very adhocly defined, and I have concerns on the validity of this proposed measurement. Related to this, if you check Table 1, you can observe that the UNet authors are using actually performed way better than other well-known baselines under this measurment, which I would assume is another clue about my concern.

---

> ### Author Rebuttal · Authors · 2025-07-31
>
> We thank the reviewer for their thorough and useful feedback. We are glad they found the paper interesting, and appreciate the number of tasks we train and evaluate on. Below, we address the reviewer’s questions on the use case, the protocols M and candidates K, the choice of baselines, and the metric used.
>
> **Use case clarification. How can we utilize the segmentation maps in the downstream or clinical tasks?**
>
> We will clarify that Pancakes is not intended to replace existing clinically validated segmentation protocols. If there is an existing tool for a particular protocol, we would advise using it. Rather, our goal is to support a new capability: enabling biomedical researchers and clinicians to explore a diverse set of plausible, semantically consistent segmentations for an unseen collection of images. After Pancakes has generated the label maps, a researcher or clinician selects which of the proposed protocols best aligns with their intended downstream use (e.g., anatomical volume analysis). We envision two broad classes of use:
>
> (1) *Rapidly segmenting new protocols*. New protocols are frequently introduced [133, 136, 137], and there is a need to produce corresponding segmentations. If a scientist has a particular protocol in mind, but there are no existing tools for segmenting it, they can choose the protocol from Pancakes that best aligns with their intended use. This choice will often suffice. In cases where Pancakes’ segmentations are not sufficiently accurate–for example, when targets are more out-of-distribution than those seen in training–the segmentation maps can offer excellent initializations to interactive segmentation systems such as ScribblePrompt [120]. To demonstrate this, we performed a preliminary experiment and found that:
>
>
> - On average, Pancakes’ predictions can be improved by 5 Dice points with a *single interactive click*, for set sizes larger than one.
>
> - On average, using ScribblePrompt with a Pancakes-initialized segmentation reduces the number of required interactions by half. If used alone, it takes ScribblePrompt 5 to 8 clicks to reach optimal quality predictions, while, when initialized with Pancakes’ predictions, ScribblePrompt can reach them in 3 to 4 clicks.
>
> Since  figures are not permitted in the rebuttal, we cannot provide plots for this experiment, but will include them in the manuscript.
>
> (2) *Exploratory population analysis*. Pancakes will support users in discovering or selecting segmentation strategies appropriate for their scientific or clinical questions. For example, a clinical scientist who studies how anatomy relates to some outcome (e.g., progression of a disease) or predictor (e.g., genetics) can use Pancakes to quickly extract segmentations of multiple candidate anatomical regions that have never been segmented, compute their volumes, and test correlations with clinical outcomes thereby identifying promising candidate regions.
>
> **Is this assessment clinically verified by the corresponding specialists? Who should confirm the clinical meaning of these protocols?**
>
> Pancakes is intended for biomedical experts. We assume that as they use the tool, they can identify the mapping between a label and a known structure. To evaluate the accuracy of our predictions, we used expert-provided segmentations for the held-out datasets (unseen at training time). We did not conduct a user study in which clinical experts evaluated the distribution of protocols. Such a study would be useful, but presents design challenges and would require a substantial investment of time by independent experts.
>
> **Did you split the dataset based on images or based on patients?**
>
> We split the dataset based on patients, and ensured that there is no train/val/test cross-contamination. We will clarify this in the paper.
>
> **Surprisingly, the original UNet can perform better than other baseline approaches.**
>
> We are unsure of what the reviewer means by Dice performance Table 1, as Table 1 relates to speed at inference instead of performance on Set Dice score. We will, therefore, address both:
> - *High UNet performance*: There is strong evidence that UNet-based architectures match transformer-based architectures in terms of performance for medical image segmentation [134, 135]. We chose to demonstrate that Pancakes works with even a simple solution, which enables fast throughput and a low number of parameters helps limit overfitting, and especially task overfitting, where the model remembers the structures to segment instead of proposing diverse segmentations. We also highlight that our proposed framework is not dependent on the choice of the model $f_{\theta_f}(.)$ that infers the distribution parameters $\phi$. The UNet implementation of $f_{\theta_f}$ could be replaced with another network, such as the SAM featurizer. We will discuss this in the revised manuscript.
> - *High Set Dice*: Since our method tackles a new task, other models are not designed to perform well on the aspects captured by these metrics (accuracy and consistency). We find that when just looking at accuracy, Pancakes is comparable to SAM and outperforms the other baselines. However, SAM cannot provide semantically consistent segmentations, which leads to substantial performance degradation when considering several images jointly.
> - *Inference Speed*: Unlike SAM-type models in automatic mode, Pancakes remains unburdened from transformer-based architecture and heavy post-processing scripts. It can run inferences for M protocols (of K labels max) in a single forward path, which guarantees much faster inference than our baselines.
>
> **Intuition behind Set Dice**
>
> Most traditional metrics fail to capture what we care about: simultaneous accuracy and consistency. Given a known label, we want to identify if it is present in our protocols *consistently*. For example, if we identify that label *(m=3, k=5)* corresponds to a given structure in one image, we want to ensure that the label *(m=3, k=5)* corresponds to the same structure in other images from the same dataset. The proposed Set Dice captures this effect by selecting the same label map for all images in the set and taking the best performing label averaged across the set. This metric choice closely approximates real-world use-cases where the user selects a preferred label consistently for all images. We also highlight that Best-Candidate Dice metric–which assumes that a user identifies the most accurate label–is frequent in the medical image literature, especially when labels are uncertain [9, 57, 83].
>
> **Additional metrics**
>
> - Note that the value for set size S=1 in Figure 8 corresponds to the traditional Dice score metric.
>
> - Following the reviewer's feedback, we included additional metrics. In addition to Set Dice, we now include Set Intersection over Union (IoU) and set surface distance (percentile 95%),
>
> *Experimental results.*
> We find that for set surface distance, Pancakes outperforms all the baselines. For set IoU, Pancakes performs similarly to SAM for S=1 (looking at a single image, equivalent to the traditional IoU metric) and outperforms the baselines in all the other cases.
> | Surface Distance          | Set Size = 1     | Set Size = 2     | Set Size = 3     | Set Size = 5     |
> |----------------|------------------|------------------|------------------|------------------|
> | MedSAM         | $16.5 \pm 2.0$   | $24.2 \pm 2.5$   | $30.3 \pm 2.3$   | $36.7 \pm 1.9$   |
> | ScribblePrompt | $21.5 \pm 8.5$   | $26.9 \pm 6.2$   | $31.9 \pm 6.0$   | $34.4 \pm 4.1$   |
> | SAM            | $15.9 \pm 6.5$   | $26.9 \pm 3.2$   | $32.6 \pm 2.3$   | $37.6 \pm 1.9$   |
> | Pancakes       | $8.2 \pm 1.3$    | $10.6 \pm 1.8$   | $12.0 \pm 2.1$   | $13.0 \pm 2.5$   |
>
> | IoU          | Set Size = 1     | Set Size = 2     | Set Size = 3     | Set Size = 5     |
> |----------------|------------------|------------------|------------------|------------------|
> | MedSAM         | $15.7 \pm 5.5$   | $11.4 \pm 4.0$   | $9.9 \pm 3.7$    | $8.3 \pm 3.5$    |
> | ScribblePrompt | $43.2 \pm 7.7$   | $30.1 \pm 5.1$   | $27.1 \pm 4.4$   | $23.4 \pm 3.4$   |
> | SAM            | $52.0 \pm 7.0$   | $33.9 \pm 4.0$   | $27.5 \pm 3.0$   | $22.1 \pm 2.7$   |
> | Pancakes       | $51.9 \pm 6.1$   | $47.8 \pm 6.5$   | $48.8 \pm 6.2$   | $49.7 \pm 5.8$   |
>
>
>
> [133]: Merdietio Boedi, Rizky, et al. "_3D segmentation of dental crown for volumetric age estimation with CBCT imaging._" International Journal of Legal Medicine 137.1 (2023): 123-130.
>
> [134]: Isensee, F. et al. (2024). _nnU-Net Revisited: A Call for Rigorous Validation in 3D Medical Image Segmentation._  In: Linguraru, M.G., et al. Medical Image Computing and Computer Assisted Intervention – MICCAI 2024. MICCAI 2024. Lecture Notes in Computer Science, vol 15009. Springer, Cham.
>
> [135]: Carsen Stringer and Marius Pachitariu. _Transformers do not outperform cellpose._ bioRxiv, pp. 2024–04, 2024
>
> [136]: _TopBrain 2025 Grand Challenge_
>
> [137]: Greve, Douglas N., et al. "_A deep learning toolbox for automatic segmentation of subcortical limbic structures from MRI images._" Neuroimage 244 (2021): 118610.

---

> > ### Comment · Reviewer_jNMX · 2025-08-08
> >
> > Thanks the authors for resolving my comments. I have no more comments now, and rating raised.

---

> > > ### Author Response · Authors · 2025-08-08
> > >
> > > We are glad we could address your comments. Should any question or concern arise before the end of the discussion period, don’t hesitate to mention it here. We thank the reviewer again for your thorough feedback. Thank you also for your careful consideration of our rebuttal and for raising your score.

---

### Official Review · Reviewer_nZep · 2025-07-02

**Clarity:** 4
**Significance:** 3
**Originality:** 4
**Rating:** 5
**Confidence:** 4

**Summary:**

The paper introduces a novel approach to biomedical image segmentation. Instead of requiring users to define a specific segmentation protocol, Pancakes automatically generates multiple complete and semantically consistent segmentation maps, each representing a plausible protocol across similar images.
Key contributions include a sampling mechanism to generate diverse segmentations from a learned distribution, a loss function that enforces semantic consistency across images within the same domain, and a lightweight and efficient model that generalizes well to unseen domains, outperforming existing methods such as SAM and MedSAM.

**Questions:**

- While the model produces diverse outputs, could there be a mechanism (e.g., visual summaries, textual prompts, or clustering) to help users choose the most relevant protocol?
- Involving clinicians or experts to judge the plausibility or utility of the predicted protocols would greatly strengthen the practical value of the model.

**Ethical Concerns:**

["NO or VERY MINOR ethics concerns only"]

**Final Justification:**

I liked this paper from the beginning. It is well-made, clearly justified, with interesting results and applicability.

**Limitations:**

yes

**Quality:**

3

**Strengths And Weaknesses:**

Strengths:
- The central idea of ensuring that the same label within a given protocol refers to the same anatomical structure across images in a dataset;
- Strong generalization to unseen domains and datasets;
- The novel loss function that rewards the most accurate prediction while encouraging consistency across samples within the same domain.
- While the concept of generating multiple segmentations per image has been explored in the literature, Pancakes is the first to explicitly enforce semantic consistency across these segmentation protocols. This opens up new directions for developing segmentation models with that on mind.

Weaknesses:
- Difficulty in assessing how many of the generated protocols are truly meaningful due to the "best-match" evaluation strategy
- From my understanding, although Pancakes produces multiple segmentation protocols, users cannot directly specify or guide the type of segmentation they want. This can limit its applicability in certain workflows.

---

> ### Author Rebuttal · Authors · 2025-07-31
>
> We thank the reviewer for their thorough and useful feedback. We are glad they found the task novel and appreciated the strong performance and efficiency of our model, and found the paper clear. We carefully address the remaining questions on the protocols and the choice of baselines.
>
> **Could there be a mechanism (e.g., visual summaries, textual prompts, or clustering) to help users choose the most relevant protocol?**
>
> Pancakes produces multiple label maps that are consistent for a set of images to segment. Currently, if the user cares about a specific region, they can choose the most relevant protocol by clicking on that corresponding label map prediction for one of the images. That protocol gets used for all the images in the set.
>
> *User Interface 1*: During the rebuttal phase, we built a simple user interface that contains 2 slider bars (one for M and one for K). As the user varies them, the segmentations proposed by Pancakes can adjust in nearly real time (enabled by our lightweight architecture) and propose consistent protocols for all the elements in the set.
>
> *User Interface 2 -- Going further*: If the user is interested in a specific region, we have designed an alternative UI where the user can specify a region they are particularly interested in by clicking, and only the labels for this specific region will be displayed.
>
> The rebuttal guidelines prohibit links, but we will make a Jupyter Notebook notebook supporting this available with the final paper to demonstrate the ease of use.
>
> While we have not explored multimodal prompts, we agree that including visual summaries and text prompts is an interesting opportunity for future research.
>
>
> **How can users directly specify or guide the type of segmentation they want**
>
> Beyond the strategies mentioned above, choosing fewer labels per map (lower K) will tend to produce coarser labels. Figure 6, Figure 7 and Figure 10 (in the supplemental material) provide an intuition of how the label maps  predicted change with M and K. During the rebuttal period, we built a simple user interface to demonstrate this effect, containing 2 sliders (for M and K). As the user adjusts them, the segmentations proposed by Pancakes adjusts in near real time (possible due to our lightweight architecture). Once M and K values Pancakes segments consistent segmentations on all images in the set. The desired protocol m or label (m, k) can then be selected through a simple click for a single image. While the NeurIPS change in rebuttal policy to disallow pdfs prohibits us from showing the interface, we will show it in the final manuscript.
>
> **Has a clinical expert evaluated the protocols?**
>
> To evaluate the accuracy of our predictions, we used expert-provided segmentations for the held-out datasets (unseen at training time). We did not conduct a user study in which clinical experts evaluated the distribution of protocols. Such a study would be useful, but presents design challenges and would require a substantial investment of time by independent experts.
>
> **How many of the protocols are useful?**
>
> This is an excellent question. To simulate this, we evaluate Pancakes on a subset of Total Segmentator, a held-out dataset that contains segmentations for many structures, with maximum 8 protocols (i.e., M = 8 ). We find that for each set, all 8 protocols would have some label maps that serve as the best candidate for some ground truth labels, meaning all protocols are used if we consider a holistic collection of possible target labels.

---

> > ### Comment · Reviewer_nZep · 2025-08-05
> >
> > Thank you for your reply. I'm satisfied with the answers.

---

> > > ### Author Response · Authors · 2025-08-07
> > >
> > > Thank you very much for your response. We are glad we could address your questions and really appreciate your feedback.

---

### Official Review · Reviewer_1J1k · 2025-07-03

**Clarity:** 4
**Significance:** 3
**Originality:** 3
**Rating:** 5
**Confidence:** 4

**Summary:**

This paper addresses an important yet under-explored area in medical imaging, particularly where foundation models are concerned, that being the lack of automated segmentation models capable of offering segmentation predictions across multiple segmentation protocols. The authors propose a method, dubbed Pancakes, that takes an image as input and outputs a distribution over segmentation protocols. Their proposed segmentation sampling mechanism then produces several complete, semantically-consistent, multi-label segmentation maps from diverse protocols for that image.

**Questions:**

* In Figure 2, what are the three methods? Are they automatic segmentation foundation models (if so, which ones?) or are they the three protocols highlighted in Figure 1?

* Did the authors evaluate the impact that the inclusion of synthetic data had on their proposed method? It would be interesting to see how stark the performance drop off is if Pancakes had to rely solely on the real medical imaging data.

* As shown in Figure 5, in the majority of cases, the second-best model is SAM. This is interesting, as SAM is not pre-trained specifically on medical images, yet as shown in the figure, SAM often outperforms the baselines that were. What is the authors’ hypothesis for this?

* On page 8, the authors state, “For our main evaluation, we use Set Dice of S = 3, and use M = 8 and K = 20 for Pancakes. The values were chosen based on our guess for typical values that would be used by a biomedical researcher who wishes to get a general and coherent understand of a set of related images”. Can the authors please elaborate on why these particular values were chosen as the “guess”? It is interesting that these same values demonstrate the best performance as shown in Figure 9. So, were these values truly a guess, or did the results shown in Figure 9 influence the selection of these values?

* In section 6, the authors highlight that in their work, they assumed “The user has an idea of the labels to segment and can select a priori a few K values”. Could the authors elaborate how this requirement of the user compares to the requirement of a user to provide a hierarchy of protocols to existing multi-protocol segmentation models as discussed in the Related Work section?

**Ethical Concerns:**

["NO or VERY MINOR ethics concerns only"]

**Final Justification:**

I thank the authors for effectively addressing most of the comments. I am happy to raise my rating from 4 to 5. I suggest the authors to incorporate the discussion points and clarifications into the main text.

**Limitations:**

yes

**Paper Formatting Concerns:**

No major formatting issues were found in this paper.

**Quality:**

3

**Strengths And Weaknesses:**

Strengths:

* The approach is well-motivated, as automated multi-protocol segmentation models are woefully underexplored in the literature.
* The authors’ proposed approach seems fairly simple and easy-to-understand, meaning that replication should be fairly straightforward.
* The proposed Pancakes model demonstrates strong performance against relevant automated segmentation foundation models, in both accuracy and semantic consistency.
* This improvement in performance also comes with a significant reduction in the number of parameters and inference time, making Pancakes much more efficient compared to existing methods.


$~$

Weaknesses:

* The performance evaluation of Pancakes seems to be lacking in robustness for the scope of the claims the authors are making. In Figures 5 and 8, Pancakes is compared against just four segmentation baselines, of which only two are pre-trained using specifically medical data. In their Related Work section, the authors discuss universal segmentation models (notably [45, 55, 98]) and several existing multi-protocol segmentation methods [19, 73, 96, 107], yet they do not compare quantitatively assess the performance of Pancakes against any of these models. Granted, the task the authors’ are exploring appears novel, but more elaboration as to why these existing universal and multi-protocol segmentation methods couldn’t be adapted in a manner where Pancakes could be compared against them.
* Other than qualitative assessment, the authors only evaluate the performance of Pancake on one quantitative metric: their novel Set Dice. While the authors do highlight the limitations of the traditional Dice metric for their specific task, relying on just a single quantitative metric of segmentation accuracy leaves questions regarding whether the strong performance of Pancakes is truly robust against various measures of segmentation quality. More discussion on the limitations of other traditional metrics for this task would have been appreciated, and the evaluation would have been enhanced by considering at least one more quantitative metric.
* While the performance improvements offered by Pancakes are sufficient enough for this not to be a major problem, it would have perhaps been better to represent the results shown in Figure 5 as a table as opposed to a bar chart, as distinguishing performance can be tricky, particularly if two models perform similarly to each other.
* Some figures that would have enhanced the comparison of Pancakes to the baseline models, such as a figure showing a qualitative segmentation comparison between them, are missing.

---

> ### Author Rebuttal · Authors · 2025-07-30
>
> We thank the reviewer for their thorough and useful feedback. We are glad they found the task novel, and appreciated our training and evaluation framework and Pancakes’ strong performance. We are also pleased that they found the paper well-motivated and clear. Below, we address the reviewers questions about  the protocols M and candidates K, the choice of baselines, and the metric used.
>
> **Limitations of traditional metrics.**
>
> A distinguishing attribute of our metric is that it takes into account the consistency of the labels produced across images of the same set. This aspect is not taken into account by other metrics. This means that the same structure (for example, a ventricle) could be classified satisfactorily, but be assigned different labels. Because structures are very rarely aligned, identifying the matching labels across different images is a challenge that is ignored with the common metrics. We will clarify this point in the paper.
>
> **Additional quantitative metrics.**
>
> - Note that the value for set size S=1 in Figure 8 corresponds to the traditional Dice score metric.
>
> - As the reviewer suggests, we included additional metrics. In addition to Set Dice, we now include Set Intersection over Union (IoU) and Set Surface Distance (percentile 95%),
>
>
> _Results_
>
> We find that for Set Surface Distance, Pancakes outperforms all the baselines. For Set IoU, Pancakes performs similarly to SAM for S=1 (looking at a single image, equivalent to the traditional IoU metric) and outperforms the baselines in all the other cases.
>
> | Surface Distance          | Set Size = 1     | Set Size = 2     | Set Size = 3     | Set Size = 5     |
> |----------------|------------------|------------------|------------------|------------------|
> | MedSAM         | $16.5 \pm 2.0$   | $24.2 \pm 2.5$   | $30.3 \pm 2.3$   | $36.7 \pm 1.9$   |
> | ScribblePrompt | $21.5 \pm 8.5$   | $26.9 \pm 6.2$   | $31.9 \pm 6.0$   | $34.4 \pm 4.1$   |
> | SAM            | $15.9 \pm 6.5$   | $26.9 \pm 3.2$   | $32.6 \pm 2.3$   | $37.6 \pm 1.9$   |
> | Pancakes       | $8.2 \pm 1.3$    | $10.6 \pm 1.8$   | $12.0 \pm 2.1$   | $13.0 \pm 2.5$   |
>
> | IoU          | Set Size = 1     | Set Size = 2     | Set Size = 3     | Set Size = 5     |
> |----------------|------------------|------------------|------------------|------------------|
> | MedSAM         | $15.7 \pm 5.5$   | $11.4 \pm 4.0$   | $9.9 \pm 3.7$    | $8.3 \pm 3.5$    |
> | ScribblePrompt | $43.2 \pm 7.7$   | $30.1 \pm 5.1$   | $27.1 \pm 4.4$   | $23.4 \pm 3.4$   |
> | SAM            | $52.0 \pm 7.0$   | $33.9 \pm 4.0$   | $27.5 \pm 3.0$   | $22.1 \pm 2.7$   |
> | Pancakes       | $51.9 \pm 6.1$   | $47.8 \pm 6.5$   | $48.8 \pm 6.2$   | $49.7 \pm 5.8$   |
>
> **Qualitative visualizations for the baselines.**
>
> Since figures are not permitted in the rebuttal, we cannot provide additional qualitative visualizations. Figure 2 shows examples of the segmentations produced by the baselines (left to right, SAM with 2 different hyperparameter choices and UnSAM). The labels produced are plausible but unlike the Pancakes labels, these labels are not consistent across images. Moreover, they are not organized, many of them overlap or some pixels are left without a label assignment. These findings are consistent across datasets and we will include them in the manuscript as well as the corresponding visualizations.
>
> **In Figure 2, what are the three methods?**
>
> These are all automatic segmentation foundation models. Left to right, we include SAM for different hyperparameter values in automatic mode and unSAM in automatic mode. In our evaluation, we select different hyperparameter values in an attempt to collect as many valid masks as possible.
>
> **Impact of the synthetic data.**
>
> We will add  an ablation study comparing training Pancakes with real data only, synthetic data only, or both. When trained with both real and synthetic data, Pancakes consistently outperforms the other variants in Dice score, (p<0.05, paired Student-t test), for M=16:
> | Data        | Set Size = 1     | Set Size = 2     | Set Size = 3     | Set Size = 5     |
> |--------------|------------------|------------------|------------------|------------------|
> | Both         | $73.2 \pm 5.5$   | $67.3 \pm 6.0$   | $67.4 \pm 6.0$   | $68.4 \pm 5.6$   |
> | Megamedical  | $71.1 \pm 6.2$   | $65.8 \pm 6.7$   | $65.7 \pm 6.8$   | $67.4 \pm 6.9$   |
> | Synthetic    | $56.3 \pm 4.9$   | $45.8 \pm 5.0$   | $44.3 \pm 5.4$   | $42.7 \pm 5.5$   |
>
>
> **Why is SAM a consistent second while not trained on medical data?**
>
> This is an excellent question, for which we don’t have a definitive answer though similar behavior has been observed in the literature[120]. We hypothesize that SAM is better because it was trained on a wider variety of labels and images. Therefore, SAM is less prone to task-overfitting compared to other models. SAM might also be more adapted to the type of interaction used to produce the predictions: clicks. We will make this point in the paper.
>
> **How were M and K selected for the main results?**
>
> We will clarify our choices of M=8 (number of maps) and K=20 (number of candidate labels) in the main paper. As Appendix Figure 15 shows, performance on the development datasets saturates at M=8 and K=20, with only marginal gains achieved by further raising M. We therefore chose to use these parameters as a balance between performance and the number of structures clinical users might actually expect in practice. Importantly, these parameters were chosen independently of the held-out test datasets in Figure 12.
>
> **How does specifying M and K compare to providing a hierarchy of protocols to existing multi-protocol segmentation models?**
>
> Specifying M and K is faster than defining a hierarchy of protocols. To demonstrate this, we have built a simple user interface that contains 2 slider bars (one for M and one for K). As the user varies them, the segmentations proposed by Pancakes can adjust in nearly real time (enabled by our lightweight architecture) and propose consistent protocols for all the elements in the set. A protocol can be selected through simple click. Since figures are not permitted in the rebuttal, we cannot share a figure showing this interface, but we will release it with the paper. This alleviates the need to manually annotate many labels and define a hierarchy a priori, especially when there is no clear hierarchy between labels.
>
> **Clarification on comparison to other universal and multi-protocol segmentation methods**
>
> We compare our model to [45] and [98] in automatic segmentation mode. Although we cite as much relevant work as possible, most are not directly applicable to the new problem setting that we tackle. For example, baselines that are data-specific or that require an explicit hierarchy are not applicable because of their limitations or heavy annotation requirements. As is often the case in medical imaging, we assume that only binary labels are available in our data. Similarly, baselines that  require multiple interactions (clicks, bounding boxes, scribbles) for each image to be segmented  are not applicable to our problem because our goal is to alleviate the burden on the user. We will clarify this choice of baselines in the revised manuscript.
>
> **Changing Bar plot to Table**
>
> Thank you for pointing this out, we will include a table for a clearer distinction between the different methods.

---

> > ### Comment · Reviewer_1J1k · 2025-08-08
> >
> > I thank the authors for their detailed response. Most of my comments have been resolved.

---

> > > ### Author Response · Authors · 2025-08-08
> > >
> > > We are glad we could address the reviewer’s concern and thank them again for helping us increase the quality of our submission. Please don’t hesitate to comment if anything remains unclear or unresolved. Thank you again.

---

### Official Review · Reviewer_kGwA · 2025-07-04

**Clarity:** 2
**Significance:** 3
**Originality:** 2
**Rating:** 3
**Confidence:** 4

**Summary:**

There are many ways to segment a given biomedical image: Segmenting regions, vessels, detecting tumors etc. Most models are trained on only a single segmentation task, or require prompting and can be inconsistent between samples. The authors introduce "pancakes", a model and a training scheme that learns to segment up to K classes on M different tasks at the same time. Rather than being preset in advance, these segmentation modalities are learned automatically from training data by attempting to learn a segmentation classes that generalize and remain consistent between many datasets and samples. The approach is evaluated on a variety of datasets and input modalities (CT, MRI etc.) and compared with other universal models such as MedSAM and UnSAM, which it outperforms.

**Questions:**

I address my questions in the weaknesses section. Right now, while I appreciate the task and contribution, the many details that are unclear me to prevent me from recommending acceptance with a clear conscience. If these become clear after the rebuttal, I would gladly reassess my score. I thank the authors in advance for their time.

**Ethical Concerns:**

["NO or VERY MINOR ethics concerns only"]

**Final Justification:**

I appreciated the authors' responses, but was still not convinced about some of the points as stated in my comments to the authors. In the end, I raised my score from 2 to 3.

**Limitations:**

The authors address limitations in their paper. I do not believe that there are further concerns about ethics or societal impact.

**Paper Formatting Concerns:**

None that I noted.

**Quality:**

3

**Strengths And Weaknesses:**

Strengths

* The task of discovering and reproducing labels consistent across many datasets and segmentation tasks rather than training for a handful of preset labels is an interesting one.
* The core parts of the approach are explained clearly, and the metrics for evaluating label consistency across tasks (Set Dice) make sense.
* The training and evaluation is solid, and comparisons to more "universal" models that could be used for the same task are favorable.

Weaknesses

* Some parts of the training are unclear to me. According to line 130-132, all protocols and labels for a sampled M, K are predicted at every training iteration. Then, according to Eq. 7, at each iteration, only the protocol-label pair that has the best prediction is optimized to encourage consistency. How is label leak across protocols prevented? Example: Suppose a dataset has Anterior and Posterior segmentations of the hippocampus as labels. What prevents the Anterior label from ending up in Protocol 0 and Posterior in Protocol 1? How do we make sure related tasks end up in the same protocol? As far as I understand, each labeling task ends up in a random protocol/label slot, which I am sure must be a misunderstanding.
* The use case for such a model is also a bit unclear. When running Pancakes on a new dataset for exploratory purposes, how can a user know which labels are what, and which protocols make sense? It is possible that some of the learned protocols make no sense on a given dataset, and in this case the "prompting" of a more universal model towards a specific even sounds like an advantage rather than a disadvantage. How would seeing a number of segmentations on a set of protocols and labels that are not well defined be helpful? I think it would be nice to clarify this with some clear real-world scenarios rather than only testing on a scenario like "S = 3, M = 8 and K = 20 sounds good" which seems intuitive but not based on a concrete use case.
* Evaluation against single-protocol models is unavailable. While the universality of the approach does indeed call for comparison to universal methods, it seems straightforward to expect that Pancakes would outperform them since they are not trained for consistency. It would be interesting to know how well the model performs against run-of-the-mill segmentation models. For example, if I want to focus on three specific protocols, what would be the advantage of using Pancakes rather than 3 separate models trained specifically on these protocols? Is the generality worth the performance trade-off? Or, does the generality and multi-model training actually improve performance? This is valuable knowledge.
* Fast inference time is put forward as an advantage of Pancakes, having only 0.22M parameters. This comes as a surprise in our current era of larger and larger models. Since this is put forward as an advantage, I think it worth exploring. Why was a small model chosen specifically? What happens with a number of parameters comparable to other models? It seems surprising that learning across large amounts of data can be this successful with such a small number of parameters at a glance.
* While the task introduced and being evaluated on is novel, most of the technical components are not. As an example, the loss being optimized on the best predicted mask at every existing iteration is done in SAM (this is also stated in the paper).
* The Limitations section states that Pancakes is strong in a resource-constrained setting, but intuitively it seems like it would do better with a larger number of datasets due to the consistency enforced between them. What is the argument for this?

---

> ### Author Rebuttal · Authors · 2025-07-30
>
> We thank the reviewer for their thorough and useful feedback. We are glad they found the task novel, and appreciated our training and evaluation framework and Pancake’s strong performance. We are also pleased that they found the paper interesting and clear. Below, we address the remaining questions on the protocol use, method applications, baseline selection, and the evaluation setting.
>
> **Could labels leak across protocols?**
>
> The primary objective of Pancakes is to propose a diverse set of consistent protocols for an image group. We do not enforce constraints across protocols produced, but we agree that at times it would be good to do so. For example, symmetric labels (left and right ventricles or posterior and anterior hippocampus) in the same protocol may be desirable. One way to  incorporate this into the Pancakes framework is to apply the loss function to two randomly chosen segmentation targets in a multi-label dataset, rather than one. To validate this, we ran an experiment on the OASIS brain dataset (2 protocols with 5 and 25 labels respectively) during the rebuttal period, and found it effective. We thank the reviewer for raising this important point and will include the method and results in the revised manuscript.
>
> **Use case clarification.**
>
> We will clarify that Pancakes is not intended to replace existing clinically validated segmentation protocols. If there is an existing tool for a particular protocol, we would advise using it. Rather, our goal is to support a new capability: enabling biomedical researchers and clinicians to explore a diverse set of plausible, semantically consistent segmentations for an unseen collection of images. After Pancakes has generated the label maps, a researcher or clinician selects which of the proposed protocols best aligns with their intended downstream use (e.g., anatomical volume analysis). We envision two broad classes of use:
>
> (1) *Rapidly segmenting new protocols*. New protocols are frequently introduced [133, 136, 137], and there is a need to produce corresponding segmentations. If a scientist has a particular protocol in mind, but there are no existing tools for segmenting it, they can choose the protocol from Pancakes that best aligns with their intended use. This choice will often suffice. In cases where Pancakes’ segmentations are not sufficiently accurate–for example, when targets are more out-of-distribution than those seen in training–the segmentation maps can offer excellent initializations to interactive segmentation systems such as ScribblePrompt [120]. To demonstrate this, we performed a preliminary experiment and found that:
>
> - On average, Pancakes’ predictions can be improved by 5 Dice points with a *single interactive click*, for set sizes larger than one.
>
> - On average, using ScribblePrompt with a Pancakes-initialized segmentation reduces the number of required interactions by half. If used alone, it takes ScribblePrompt 5 to 8 clicks to reach optimal quality predictions, while, when initialized with Pancakes’ predictions, ScribblePrompt can reach them in 3 to 4 clicks.
>
> Since  figures are not permitted in the rebuttal, we cannot provide plots for this experiment, but will include them in the manuscript.
>
> (2)  *Exploratory population analysis*. Pancakes will support users in discovering or selecting segmentation strategies appropriate for their scientific or clinical questions. For example, a clinical scientist who studies how anatomy relates to some outcome (e.g., progression of a disease) or predictor (e.g., genetics) can use Pancakes to quickly extract segmentations of multiple candidate anatomical regions that have never been segmented, compute their volumes, and test correlations with clinical outcomes thereby identifying promising candidate regions.
>
> **How can a user identify structures and know which protocols make sense?**
>
> Pancakes is intended for biomedical experts. We assume that as they use the tool, they can identify the mapping between a label and a known structure. Because Pancakes provides consistent segmentations for groups of images, this provides some context on what labels represent. In future work, we plan to investigate automating the mapping between a structure and known protocols.
>
>
> **What would be the advantage of using Pancakes rather than 3 separate models trained specifically on these protocols?**
>
> Pancakes is not intended to replace clinically validated segmentation protocols. If the user knows which targets they wish to segment, and good models exist for those targets, those models should be used. Pancakes is meant for scenarios where the targets are unknown or for which there are no existing segmented scans or segmentation models.
>
> **While the task introduced and being evaluated on is novel, most of the technical components are not. As an example, the loss being optimized on the best predicted mask at every existing iteration is done in SAM (this is also stated in the paper).**
>
> We agree that the individual components of our framework have been explored in prior contexts. Our key contribution is in formulating an interesting new problem, and in carefully adapting and integrating techniques to solve it. This includes developing a version of the loss function that supports consistency, adapting the synthetic data so that the produced samples are consistent and proposing a lightweight architecture that can efficiently sample protocols. However, we emphasize that our main contribution lies in introducing a fundamentally new framing to medical image segmentation, and alleviating the annotation requirements on the user.
>
> **Small model and resource constrained setting**
>
> There is strong evidence that UNet-based architectures achieve performance similar to  transformer-based architectures in medical image segmentation [134, 135]. Using a relatively small number of parameters limits overfitting, and especially task-overfitting, where the model remembers the structures to segment instead of proposing diverse segmentations. An additional benefit of using a small model is inference time efficiency. We did not have the time to fully train larger models during the rebuttal period. However, preliminary results indicate that increasing the number of parameters does not lead to an improvement in performance. We also highlight that our proposed framework is not dependent on the choice of the neural network that captures the function $f_{\theta_f}(.)$, which infers the distribution parameters $\phi$. The UNet implementation of $f_{\theta_f}$ could be replaced with another network, such as the SAM featurizer. We will discuss this in the revised manuscript.
>
> [133]: Merdietio Boedi, Rizky, et al. "*3D segmentation of dental crown for volumetric age estimation with CBCT imaging.*" International Journal of Legal Medicine 137.1 (2023): 123-130.
>
> [134]: Isensee, F. et al. (2024). *nnU-Net Revisited: A Call for Rigorous Validation in 3D Medical Image Segmentation*.  In: Linguraru, M.G., et al. Medical Image Computing and Computer Assisted Intervention – MICCAI 2024. MICCAI 2024. Lecture Notes in Computer Science, vol 15009. Springer, Cham.
>
> [135]: Carsen Stringer and Marius Pachitariu. *Transformers do not outperform cellpose*. bioRxiv, pp. 2024–04, 2024
>
> [136]: *TopBrain 2025 Grand Challenge*
>
> [137]: Greve, Douglas N., et al. "*A deep learning toolbox for automatic segmentation of subcortical limbic structures from MRI images.*" Neuroimage 244 (2021): 118610.

---

> > ### Comment · Reviewer_kGwA · 2025-08-08
> >
> > Many thanks to the authors for their detailed rebuttal. While I now have a better understanding of the proposed approach, I remain somewhat unconvinced about the practical use case for multi-protocol segmentation. In many biomedical contexts, one might expect an expert to already know the target classes and to use a specialized model. Pancakes could be valuable for exploratory purposes, but it may not always align with the expert’s goals and could require further refinement, for example with ScribblePrompt, as the authors note. Regarding the small model setting, I am still a bit unclear on the motivation. Although [134] highlights the effectiveness of UNets, larger models often achieve stronger results; cases where a smaller model outperforms a larger one may indicate factors such as limited dataset size or incomplete training. In short, a real use case scenario, possibly supported by the use of a real expert and further analysis of some components would go to great lengths in supporting the paper. That said, I appreciate the authors clarifying several of my points and raise my initial score.

---

> > > ### Author Response · Authors · 2025-08-08
> > >
> > > Thank you for your thoughtful review of both the paper and our rebuttal.  Thank you also for raising your score. We appreciate your push to clarify the use case and will highlight that this work stems from close collaboration with biomedical experts. While larger models have so far yielded limited gains in our experiments, we agree that further exploration is worthwhile. We believe this work is a strong fit for NeurIPS: though not yet clinically ready, it outperforms all baselines and lays a strong foundation for future methods.

---

### Note · Authors · 2025-08-13

We appreciate that the reviewers engaged seriously with our responses and are encouraged that all expressed satisfaction with the rebuttal, with two indicating they would raise their scores. Reviewers agreed on the novelty of the task, the interest of the problem, and Pancakes’ strong performance, noting that it opens up a new direction for research in medical imaging.

In response to reviewer suggestions, we strengthened the paper by adding complementary evaluation metrics (IoU, surface distance), expanding comparisons and ablations (including state-of-the-art interactive segmentation), illustrating how Pancakes can initialize interactive segmentation, and refining our discussion of goals, use cases, and limitations.
We were happy to see that all reviewers responded positively and appreciated the rebuttal, indicating that it eliminated nearly all worries.

---

### Decision · Program_Chairs · 2025-09-17

**Decision:**

Accept (poster)

**Comment:**

Authors present a method to generate multiple consistent parcellations
for a given image as well as a given image set. To this end, they
introduce a network and a random segmentation coding similar to
positional encoding. The network produces a representation that can
produce multiple parcellations and then a shallower network converts
the representation to different segmentation maps based on the random
encoding. The details of this process however, are not very clearly
described in my opinion. The sampling process, the architecture of the
final network - how is it able to output variable number of output
segmentation maps is unclear - as well as the computation of the
loss - in particular how Equation 6 and 7 enforces consistent
segmentation across samples - requires further details.

The paper's goal is to be able to segment any image and generate
multiple segmentation possibilities for the same image. These
segmentations seem to be dense - the method outputs a representation
per pixel and these are mapped to segmentations for each pixel based
on the random protocol.

The main strength of the article is the way the task is defined. The
task itself and how the segmentation is formulated is interesting
indeed.

The main weaknesses the reviewers highlight are: (1) lack of clarity
in the explanations, (2) unclear motivation and use cases, and (3)
experimental design, especially lack of comparisons with task-specific
segmentation models and introduction of a new metric.

Despite the weaknesses, reviewers seem to have valued the innovation
in the paper.

**Summary**: The article starts with an interesting and novel task
definition. Overall the method seems interesting as well. The clarity
of the descriptions are not very high quality. Authors tried to remedy
this issue in their rebuttals. The use-case of the method is not very
clear. While authors tried to remedy this issue in the rebuttal, I do
not see a clear answer. Despite the weaknesses, the reviewers value
the novelty of the paper and think this would be a good
contribution. I also think it may start some interesting discussions.